# Strong hydroclimatic controls on vulnerability to subsurface nitrate contamination across Europe

R. Kumar [1✉], F. Heße[1], P. S. C. Rao[2,3], A. Musolff [1], J. W. Jawitz [4], F. Sarrazin[1], L. Samaniego [1], J. H. Fleckenstein[1,5], O. Rakovec [1,6], S. Thober[1] & S. Attinger[1,7]

Subsurface contamination due to excessive nutrient surpluses is a persistent and widespread problem in agricultural areas across Europe. The vulnerability of a particular location to pollution from reactive solutes, such as nitrate, is determined by the interplay between hydrologic transport and biogeochemical transformations. Current studies on the controls of subsurface vulnerability do not consider the transient behaviour of transport dynamics in the root zone. Here, using state-of-the-art hydrologic simulations driven by observed hydroclimatic forcing, we demonstrate the strong spatiotemporal heterogeneity of hydrologic transport dynamics and reveal that these dynamics are primarily controlled by the hydroclimatic gradient of the aridity index across Europe. Contrasting the space-time dynamics of transport times with reactive timescales of denitrification in soil indicate that ~75% of the cultivated areas across Europe are potentially vulnerable to nitrate leaching for at least one-third of the year. We find that neglecting the transient nature of transport and reaction timescale results in a great underestimation of the extent of vulnerable regions by almost 50%. Therefore, future vulnerability and risk assessment studies must account for the transient behaviour of transport and biogeochemical transformation processes.

[1] UFZ-Helmholtz Centre for Environmental Research, Leipzig, Germany. [2] Lyles School of Civil Engineering, Purdue University, West Lafayette, IN, USA. [3] Agronomy Department, Purdue University, West Lafayette, IN, USA. [4] Soil and Water Sciences Department, University of Florida, Gainesville, FL, USA. [5] Bayreuth Center of Ecology and Environmental Research, University of Bayreuth, Leipzig, Germany. [6] Faculty of Environmental Sciences, Czech University of Life Sciences, Prague, Czech Republic. [7] University of Potsdam, Potsdam, Germany. ✉email: rohini.kumar@ufz.de

Despite >15 years of water quality protection implementation under the EU Water Framework Directive (EU-WFD[1]), the most recent EU-WFD report[2] concludes that the majority of European water bodies do not meet the European Union's minimum target, with threats coming from a wide range of pollutants. Among these, excess nitrate from agricultural areas was highlighted as a major concern[3–8]. Consequently, the European Nitrate Directive[9]—itself an integral part of the EU-WFD—designates nitrate vulnerable zones (NVZs) as areas at risk from agricultural nitrate pollution and requires prompt actions to improve nitrate management. A number of indices have been developed to delineate these zones[10–13]. While these indices differ in their conceptual and implementation modes, they are often based on a weighted combination of temporally invariant environmental parameters (e.g., terrain slope, land cover and subsurface properties, mean precipitation). A framework for the delineation of such NVZs based on an integrated understanding of the complex and dynamic interplay between hydrologic transport and biogeochemical turnover is still missing.

A major challenge for such a framework is to capture the hydrologic transport capacity or the intrinsic vulnerability to subsurface contamination by diffuse pollutants[11,14]. Subsurface transport is particularly elusive and uncertain due to its complex flow patterns. To account for this uncertainty, research has focused on the statistical characterization of transport dynamics through travel-time distributions (TTDs), which capture the journey of water and dissolved solutes through a given subsurface compartment[15–20]. Much work has been based on steady-state TTDs, however, more recently, studies have started to acknowledge the transient nature of TTDs[21–27]. Typically, such studies have focused on empirical observations at the catchment scale or at a limited number of densely gauged small-scale catchments. While transport dynamics have recently been investigated at larger-scales[28], there are, however, no studies that systematically examine the transient nature of travel times, identify the main driving forces, and connect them to the reactive behavior of (diffuse) pollutants at regional to continental scales. This information would be relevant for management and decision making.

To address this gap, we provide a Europe-wide assessment of hydrologic transport behavior as an integrated measure of the intrinsic vulnerability to subsurface contamination by diffuse pollutants (e.g., nitrate). We then use the case of widespread nitrate contamination across arable lands in Europe to show the unrecognized importance of the transient nature of hydrologic transport in previous vulnerability and risk assessments. Our analysis is based on state-of-the-art continental-scale hydrologic simulations driven by meteorological observations over the period of 1950–2015 combined with the recent theoretical developments for characterizing the transient nature of hydrologic transport dynamics[21,22,26,27] at high spatial and temporal resolutions (0.25° and daily timescale; see "Methods"). We focus on the root zone because it is the interface between the land surface and deeper subsurface. This zone is the most dynamic and active part of the subsurface and acts as both a hydrologic and a biogeochemical filter, determining the delivery and turnover of surface inputs and the partitioning of flow paths to near and deeper subsurface waters[29–31]. The rooting depth varies across European landscapes depending on, among other geophysical attributes, vegetation types and groundwater table depth[32]. However, our focus in this vulnerability assessment is limited to arable landscapes, we therefore account for the dynamics of the first meter of soil that mostly coincides with the rooting zone for arable lands[33]. We use the dimensionless Damköhler number[34–36] to link the hydrologic and biogeochemical timescales (see "Methods") and provide an objective measure for the

large-scale vulnerability assessment of subsurface nitrate contamination. Our study therefore focuses on Europe-wide vulnerability assessment, i.e., the potential for (excess) nitrate leaching from the root zone to deeper in the subsurface (i.e., vadose zone below rooting depth). We demonstrate the oversimplified (static) nature of previous vulnerability assessment approaches by highlighting the relevance of the transient nature of transport dynamics, and we discuss its ramifications for future assessment and subsequent policy decisions. Our continental-scale analysis demonstrates strong spatiotemporal heterogeneity of hydrologic transport dynamics pronounced throughout the European landscapes, and we show that the (static) vulnerability assessment approach that does not account for such transient features greatly underestimates the extent of vulnerable areas prone to subsurface contamination by excess nitrate leaching.

## Results and discussion

**Space-time variability of hydrologic transport times**. Our continental-scale hydrologic simulations show large space-time heterogeneity in the inferred TTDs, which illustrates the complex, non-linear, and transient nature of transport dynamics in the root zone (Fig. 1; see also Supplementary Video). The large spread among the simulated daily TTDs in three exemplary locations (Fig. 1a–c) illustrates the pronounced (space-time) heterogeneity of hydroclimatic factors (e.g., precipitation, soil-water storage, infiltration and evapotranspiration fluxes), and characterizes the different transport dynamics inferred across Europe. These locations represent the humid, sub-humid (transitional), and semi-arid climate regimes, aridity indices ($\phi$ = ratio between mean potential evapotranspiration and mean precipitation) of 0.25 (UK), 1.15 (France), and 3.25 (Spain), respectively. A consistent shift towards longer travel times is noticed when moving from humid to semi-arid regions. The daily travel-time interquartile range ($TT_{IQR}$) increases jointly with the median travel-time ($TT_{50}$) throughout Europe (Fig. 1d–f and see Supplementary Fig. 1). The humid location (UK; Fig. 1a–d) exhibits marked seasonality with higher $TT_{50}$ in summer, which is likely due to a regular seasonal pattern of hydrologic states/fluxes, specifically soil moisture and evapotranspiration. In contrast, the daily dynamics of the $TT_{50}$ at the semi-arid location (Spain; Fig. 1c, f) are more erratic and episodic in nature. This location is characterized by infrequent rainfall that, when combined with high evapotranspiration losses, leads to highly variable soil-water storage. The location in France marks a transition zone between humid and semi-arid locations, with a seasonal pattern during wet years and an erratic pattern during dry years (Fig. 1b, e). The distinct behavior of the TT dynamics simulated among different locations broadly agrees with past theoretical understandings, even though these previous efforts used synthetic datasets[29,37].

**Synthesis of hydrologic transport times across Europe**. Figure 2a, b summarizes the spatial patterns of the temporal mean $\mu(TT_{50})$ and standard deviation $\sigma(TT_{50})$ of the daily soil-water travel times across Europe. The ranges for both the mean and the standard deviation span factors of 6 to 7 across the continent, as 99% of the values lie between 100 and 700 days for $\mu(TT_{50})$ and 52 and 320 days for $\sigma(TT_{50})$. Fifty percent of the study domain has $\mu(TT_{50})$ and $\sigma(TT_{50})$ values that exceed 365 and 120 days, respectively. Regions with the shortest soil travel times ($\mu(TT_{50}) \leq$ 180 days) are located in areas of high and frequent rainfall (e.g., alpine, northern UK and northern Spain— Pyrenees). The longest travel times ($\mu(TT_{50}) \geq$ 540 days) are found in dry areas with less frequent rainfall in southern Spain and the eastern European regions adjacent to the Black Sea. The spatial patterns (Fig. 2a, b) suggest a strikingly high spatial similarity

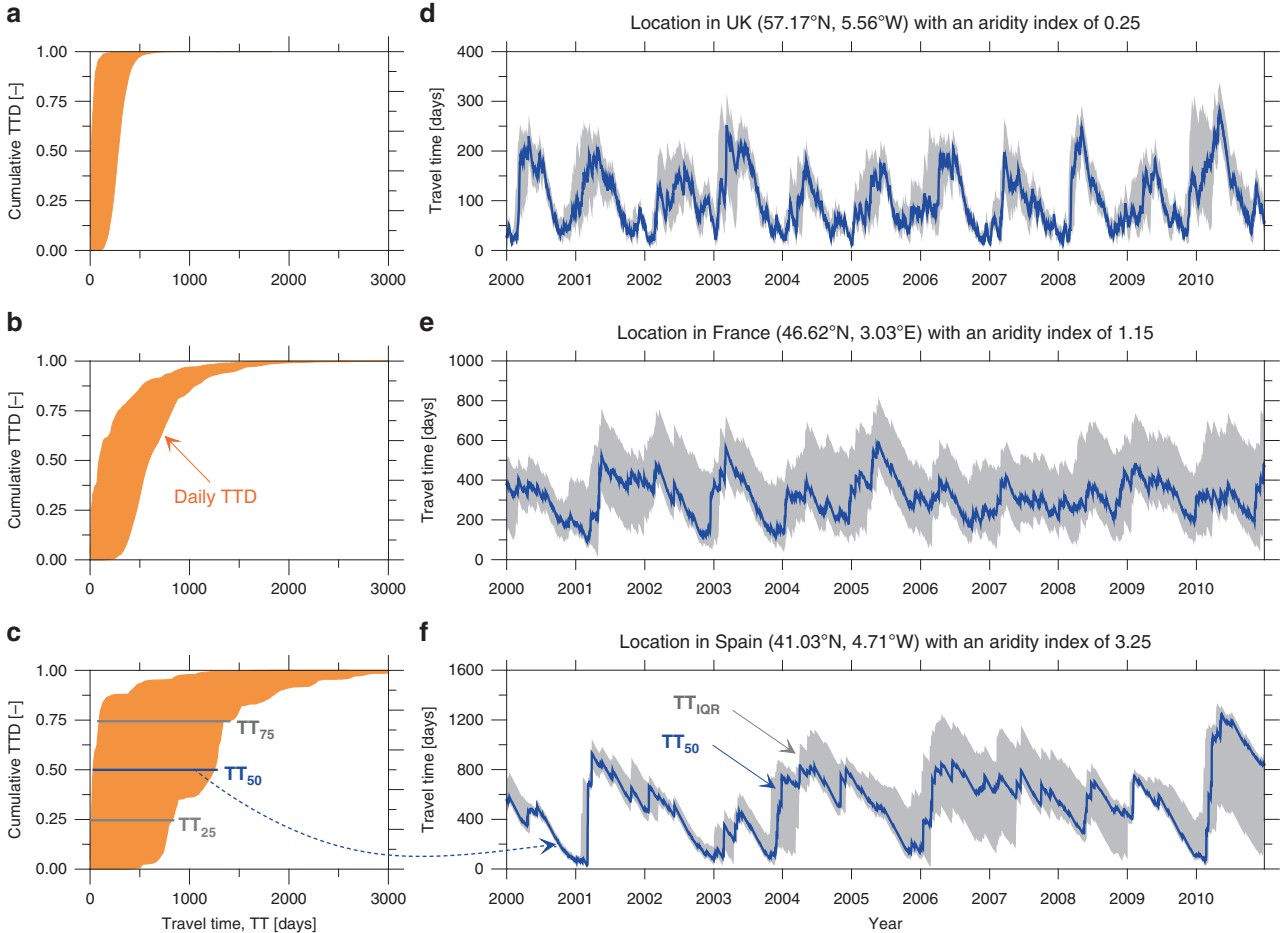

**Fig. 1 Transient features of the hydrologic transport dynamics.** Illustration of the daily travel-time distributions (**a–c**), and the corresponding temporal evolution of the daily median $TT_{50}$ and interquartile ranges $TT_{IQR}$ (**d–f**) for three distinct locations in the UK, France, and Spain representing the sub-humid, humid or transitional, and semi-arid hydroclimatic settings across Europe, respectively.

between $\mu(TT_{50})$ and $\sigma(TT_{50})$. This was confirmed through a linear regression analysis (Fig. 2d), resulting in $R^2 = 0.72$ and 0.97 for raw and binned data ($p$-value $< 0.00001$). Consequently, European regions that have, on average, longer soil travel times are also more variable or episodic in time, and vice versa. This result also means that the coefficient of variation (CV) of the daily $TT_{50}$, which is the regression slope between $\mu(TT_{50})$ and $\sigma(TT_{50})$, is remarkably consistent across Europe (~0.4). An analogus consistency in CV values has been also reported in a previous study[37] for catchment-scale travel times across different hydroclimatic settings with synthetic datasets. The temporal variability in median travel times is strongly controlled by variability in climate, and we find that the value of $TT_{50}$ CV = 0.4 is in good agreement with the temporal variability of daily potential evapotranspiration (CV = 0.34, see Supplementary Fig. 2).

The rather well-organized spatial patterns of $\mu(TT_{50})$ and $\sigma(TT_{50})$ follow the hydroclimatic gradient observed across Europe (Fig. 2c), and here, the latter pattern is represented through the aridity index ($\phi$) that primarily controls the partitioning of incoming rainfall and energy into outgoing water fluxes (i.e., evapotranspiration vs. runoff). Approximately 70–73% of the variance in the Europe-wide estimates of $\mu(TT_{50})$ and $\sigma(TT_{50})$ can be explained solely by the spatial heterogeneity of $\phi$ (Fig. 2e, f). The spatial correlation structure for these travel-time characteristics and the aridity index are nearly identical (Supplementary Fig. 3). The close relation of $\mu(TT_{50})$ and $\sigma(TT_{50})$ to $\phi$ emphasizes that the high soil moisture and frequent rainfall conditions in

humid regions lead to a high connectivity and fast displacement of water in the soil column[29]. In contrast, the (semi-)arid regions with infrequent rainfall and high soil-water deficits due to high evapotranspiration losses generally exhibit long travel times.

Other site-specific landscape attributes related to terrain, soil and vegetation characteristics show a weaker correspondence to the spatial heterogeneity of $\mu(TT_{50})$ and $\sigma(TT_{50})$ compared to $\phi$ and, therefore, constitute secondary controls (see Supplementary Fig. 4). Notably, the combined effect of individual secondary factors appears to be stable across the range of $\phi$ values, as is visible in the scatter of points in Fig. 2e, f; as well as by the overlying standard deviation estimates of the respective binned data (for every $\phi$ interval of 0.15). For the (average) binned estimates, we find an almost perfect linear correlation of $\phi$ with $\mu(TT_{50})$ and $\sigma(TT_{50})$ ($R^2 = 0.96$–0.98; $p$-value $< 0.00001$). These results are also found for the extremes of the daily TT distributions. For example, $TT_{10}$ and $TT_{90}$, which are indicative of young and old water fractions, respectively, also show a high spatial correlation with $\phi$ (see Supplementary Figs. 5 and 6). This result underpins the dominant role of the hydroclimatic factor ($\phi$) in shaping the dominant features of hydrologic transport timescales inferred across Europe.

**Vulnerability to nitrate leaching across the cultivated areas of Europe.** Thus far, our analysis has focused on the hydrologic transport dynamics that represent the intrinsic vulnerability[14] of

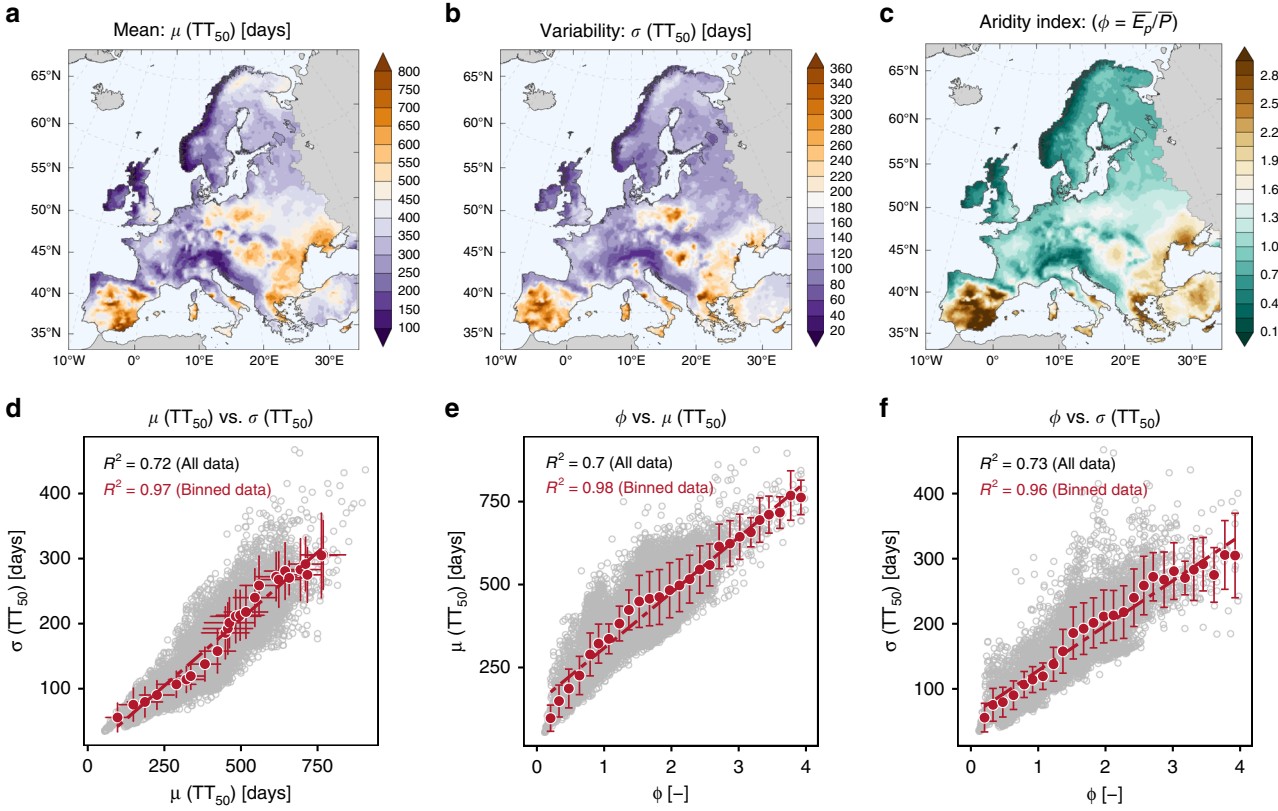

**Fig. 2 Synthesis of the hydrologic transport time dynamics across Europe.** The transient feature of the daily median travel times ($TT_{50}$; blue lines in Fig. 1d, e, f) is summarized as their temporal mean, $\mu(TT_{50})$ and standard deviation, $\sigma(TT_{50})$ (**a**, **b**) for the period 1985–2015. The strong spatial correspondence between $\mu(TT_{50})$ and $\sigma(TT_{50})$ is evident through the point-wise correlation analysis (**d**). The prevailing hydroclimatic feature in the form of aridity index, $\phi$ in (**c**), the ratio between the mean potential evapotranspiration ($\overline{E_p}$) and mean precipitation ($\overline{P}$), is identified as the dominant factor controlling the spatial heterogeneity of the transient transport characteristics as $\mu(TT_{50})$ and $\sigma(TT_{50})$ simulated across Europe (**e**, **f**). On each of the scatter plots, along with point-wise cloud data, there is also the corresponding bin estimates given as the mean and one standard deviation values of grouped data for every $\phi$ interval of 0.15. Binning is performed with the aim of seeking generalized relationships (i.e., after reducing the noise in scatter due to outliers) and specifically in (**e**, **f**) to depict the role of secondary (landscape-related) factors through the representation of the binned standard deviation estimates that are relatively stable and present across the whole range of $\phi$ values.

the system to subsurface contamination. Here, we complement the transport timescales with the biogeochemical turnover timescales of nitrate in soil to determine the extent of vulnerable regions to nitrate leaching across the cultivated areas of Europe. We focus on two competing processes of excess nitrate removal (after consideration of plant uptake) from the soil by contrasting the timescales of nitrate leaching (hydrologic transport) and denitrification (biogeochemical turnover). Denitrification rates are poorly constrained due to a lack of reliable observations, especially at large scales. To acknowledge this uncertainty, we consider different characteristic denitrification timescales[38], $\langle RT_{50} \rangle$, defined here as the 50% removal of the initial substrate, to allow comparability with the median transport times ($TT_{50}$). We consider a range of $\langle RT_{50} \rangle$ values between 0.5 and 5 years in our analysis[39–43] (see "Methods" for more details). $\langle RT_{50} \rangle$ represents the effective timescale encapsulating relevant environmental factors, such as soil moisture, temperature, and organic carbon content, that affect the site-specific reaction rates[42,44]. In the following text, we analyze two cases and conduct a nitrate vulnerability assessment corresponding to the static (time-averaged) and transient behaviors of the hydrologic transport (TT) and denitrification (RT) timescales.

We connect the transport and denitrification timescales through the dimensionless Damköhler number (here defined as $D_a = \frac{TT_{50}}{RT_{50}}$) that enables us to assess the interplay between these

two competing processes across the geographical domain[34–36,38]. When $D_a < 1$, transport (leaching) dominates over reaction (denitrification), and vice versa. Our static vulnerability assessment case, based on the range of $\langle RT_{50} \rangle$ (0.5–5 years) and the averaged transport times $\mu(TT_{50})$, results in $D_a$ values ranging between 0.05 and 4.0 across the majority of European cultivated areas (Fig. 3a). To interpret this $D_a$ number, we rely on a series of prior studies. First, a prior study[34] presents a variety of field observational datasets that demonstrate an empirical (non-linear) relationship between $D_a$ and nitrate removal. Subsequent studies[36,45] showed how this empirical relationship can be described by a parametric model based on the exponential function. Using the latter, the above value of $D_a = 0.05$ would imply more than 90% of the nitrate leaching from soil, while a value of $D_a = 4.0$ would correspond to less than 10% of the nitrate leaching from soil. Furthermore our results suggest that the majority of the cultivated areas in Europe would be highly vulnerable to nitrate leaching ($D_a \leq 1$) under the higher $\langle RT_{50} \rangle$ value ($\geq 2$ years), while very few locations would be classified as vulnerable in the case of the lowest $\langle RT_{50} \rangle$ value of 0.5 year due to the dominance of the denitrification timescale.

We further elaborate on the moderate case of an $\langle RT_{50} \rangle = 1$ year, which is a highly plausible value, as inferred by the values of denitrification timescales in previous large-scale[39,42,46] and catchment-scale studies[40,43]. Using this $\langle RT_{50} \rangle$ value and the

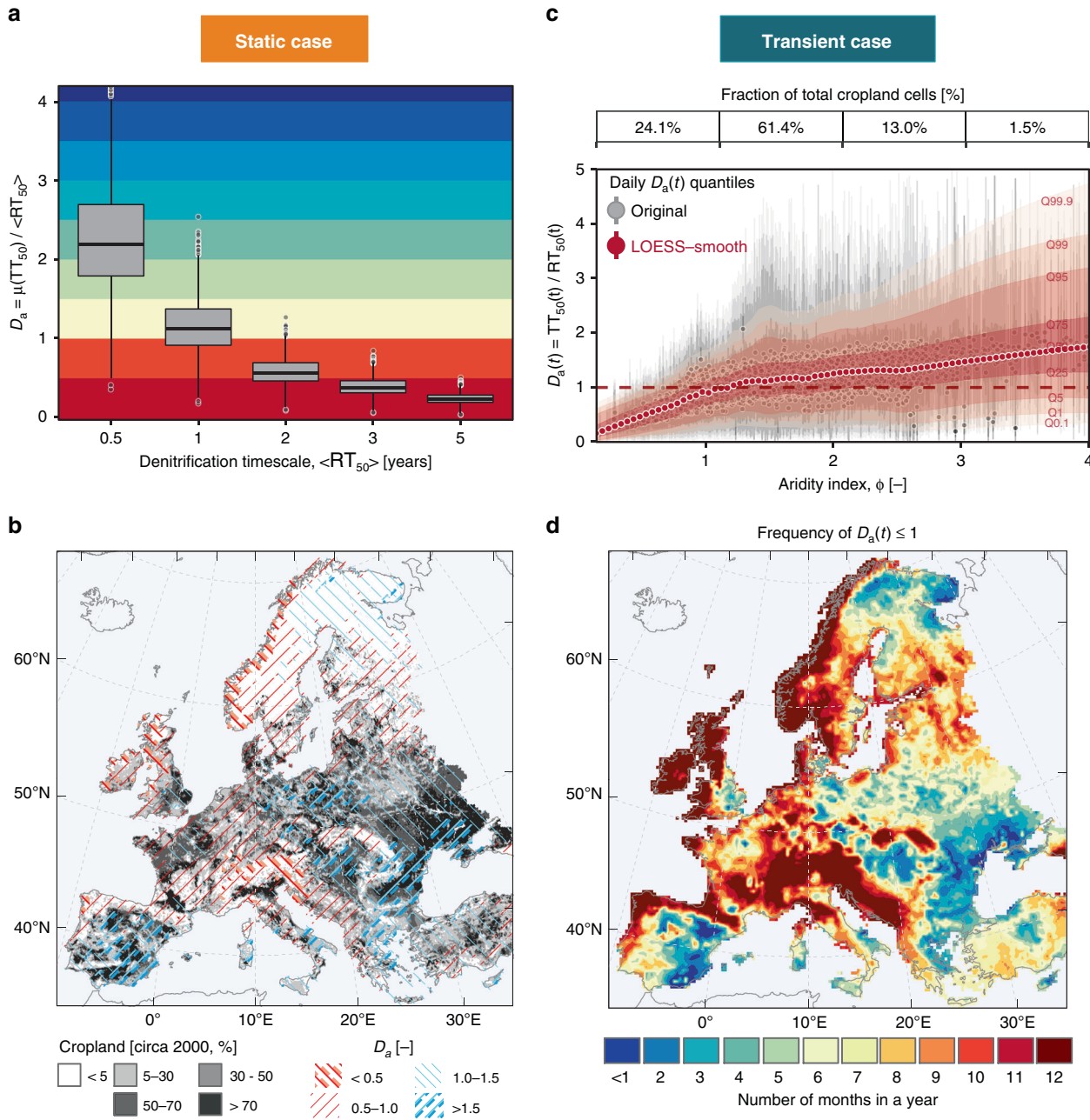

**Fig. 3 Subsurface nitrate vulnerability assessment across Europe under static and transient considerations of hydrologic transport and denitrification timescales. a** Box-plots summarizing the spatial distribution of the Damköhler number ($D_a = \frac{TT_{50}}{RT_{50}}$) for a range of effective denitrification timescales $\langle RT_{50} \rangle$ and the averaged transport times $\mu(TT_{50})$ estimated across the cultivated areas of Europe[72]. Boxplot is displayed with the horizontal bar at the median, the box indicates the first and third quartiles and the whiskers indicate ±1.5 times the interquartile range. Grid cells with at least 5% of the cropland areas are considered cultivated areas in the analysis. **b** Europe-wide $D_a$ estimates for the static vulnerability assessment case corresponding to the effective reaction timescale, $\langle RT_{50} \rangle$ and the average transport timescales, $\mu(TT_{50})$. Areas with $D_a \leq 1$ are vulnerable to subsurface nitrate leaching and ~42% of the total cultivated area falls within this category under the static vulnerability assessment case. **c** Summary of the transient daily $D_a(t)$ estimates for every cultivated grid cell, arranged according to their corresponding aridity index values ($\phi$). Summary statistics are presented as quantiles of the daily $D_a$ values (Q0.1,1,5, …,95,99,99.9) for the actual estimates (in background) and the LOESS (locally weighted smoothing)-derived smooth statistics (in a transparent foreground for better clarity). **d** Frequency of the daily $D_a(t) \leq 1$ estimated based on the time-varying $TT_{50}(t)$ and $RT_{50}(t)$. Considering the transient nature of $D_a$, nearly 75% of cultivated areas of Europe become vulnerable to nitrate leaching ($D_a(t) \leq 1$) for at least one-third of the year (i.e., frequency estimates ≥ 0.33 or 4/12).

average transport time $\mu(TT_{50})$, we find ~42% of the cultivated areas across Europe to be vulnerable to subsurface nitrate leaching, with $D_a \leq 1$ (Fig. 3b). Figure 3b shows that the cultivated areas in the humid and transitional climate zones ($\phi \leq 1.5$), specifically in the western part of Europe, i.e., France, Germany,

Italy, UK and Ireland, are dominated by hydrologic transport ($D_a \leq 1$) and are thus vulnerable to nitrate leaching. In contrast, the agricultural areas in the Iberian Peninsula and eastern European countries present higher dominance of denitrification over the hydrologic transport processes. The vulnerable areas

delineated by our approach based on the Europe-wide $D_a$ map are remarkably consistent with the (nitrate) leaching risk potential map published by the European Commission[47] and this map was established following a static indices-based approach (see Supplementary Fig. 7 for more details).

We now contrast the above static characterization of nitrate vulnerability with the results of the transient case based on the time-varying daily $TT_{50}(t)$ and $RT_{50}(t)$. Here, we account for the spatiotemporal variability of the environmental factors, $f_E(t)$, that affect the daily dynamics of the denitrification timescale, $RT_{50}(t)$. We include the effect of varying soil moisture, air temperature, and organic carbon content following established parameterization schemes[39–41,43,48] (see "Methods" for more details). This allowed us to construct the temporal variability of $RT_{50}(t)$ for each grid cell by explicitly accounting for the space/time variability of the environmental factor ($f_E$) (see "Methods"). The daily mean and variability of the environmental factor ($f_E$) follows the hydroclimatic gradient of the aridity index $\phi$ observed across Europe (see Supplementary Fig. 8). We observe lower mean and variability of the daily $f_E$ in (semi-)arid areas, resulting from the relatively drier soil moisture conditions that persist for long periods, than in humid areas that have on average higher $f_E$ because of wetter and more strongly seasonal soil moisture dynamics (Supplementary Fig. S8).

Next we analyze the variability of the daily $D_a(t)$ that depicts the interplay between the hydrologic transport timescale ($TT_{50}$) and the reactive timescale ($RT_{50}$) behaviors for the excess nitrate removal (leaching vs. denitrification) through the soil. Here, in Fig. 3c we summarize the daily $D_a(t)$ as the quantile estimates for every cropland cell arranged according to their respective aridity index ($\phi$) value. For example, for a given cropland cell located in a climate region of $\phi = 1$ and having the 50th percentile (or median; Q50) value of the daily $D_a = 1.0$, this number would indicate that the given cell would be prone to excess nitrate leaching (i.e., $D_a \leq 1.0$) for nearly half of simulation times (on average six months of a year). The results depicted in Fig. 3c clearly show the increasing range of the daily $D_a$ variability (e.g., between Q95 and Q5) when moving from humid to semi-arid and arid regions. We observe a nearly twofold (100%) increase in the $D_a$ range (Q95-Q5) for cropland cells with $\phi$ of 1 to 3. Interestingly, while the averages (medians) of the daily $D_a$ values are usually high (above 1) in (semi-) arid regions, the $D_a$ values also exhibit high temporal variability and, therefore, frequently fall below a critical value of 1 (i.e., the favorable times for (excess) nitrate leaching). The correlation analysis suggests a strong correspondence between the averaged $D_a$ statistics (median and Q95-Q5) and the aridity index ($\phi$) across the European croplands ($R^2 \geq 0.92$).

We now analyze how the temporal dynamics of the daily $D_a(t)$ affects our vulnerability assessment. The results of the frequency analysis based on the time-varying $D_a \leq 1$ suggest that ~75% of the cropland cells across Europe are vulnerable to nitrate leaching for at least one-third of the year (Fig. 3f; see also Supplementary Fig. 9). Our estimate of potentially vulnerable areas is nearly twice the estimate (42%) obtained above under the static consideration of transport and reaction timescales. Importantly, the cultivated areas located in the Iberian Peninsula and eastern European countries, which were not recognized as vulnerable regions under the static assumptions (Fig. 3b), are now regarded as regions that are temporarily vulnerable to subsurface nitrate contamination (Fig. 3c, d). We find that the majority of cropland cells in Europe (>90%) are prone to nitrate leaching for at least two months of the year (see Supplementary Fig. 9). Therefore, our results highlight the limitation of the static vulnerability assessment approach[47], which leads to an underestimation of nitrate vulnerable regions and has serious implications for nitrate management across Europe.

**Concluding remarks**. In this study, we provide a Europe-wide assessment of transient hydrologic transport behavior in the upper one meter of the subsurface. This approach allows us to quantify the intrinsic vulnerability of the subsurface to contaminant leaching at unprecedented spatial and temporal resolutions at the continental scale. We demonstrate the dominant role of large-scale hydroclimatic factors, as expressed in the aridity index, in determining the spatial heterogeneity of transport characteristics (e.g., temporal mean and variability) and the environmental factors that affect the daily variability of denitrification timescales. We apply an approach based on the dimensionless Damköhler number to characterize the vulnerability of subsurface waters to (excess) nitrate leaching from soil by accounting for the complex and dynamic interplay between the hydrologic transport and biogeochemical (denitrification) reaction timescales. This approach provides a general framework to objectively assess the vulnerability to other agrochemical pollutants in different subsurface compartments (e.g., root zone, deeper vadose zone, and eventually shallow and deep groundwater).

Using this framework as a decision tool for subsurface nitrate contamination assessments, our study closes a pressing gap by making the recent progress in the field of transport dynamics accessible to practitioners, regulators, and decision-makers that aim to safeguard and restore European waters. Our results/ framework can be used to identify hot spots and times vulnerable to (excess) nitrate leaching through the soil, and thereby could be used for assisting (nitrate) management strategies, such as the optimization/regulation of fertilizer applications. Our emphasis on the transient aspects in the process of defining vulnerable zones will become more important due to the projected increases in the frequency and intensity of extreme hydroclimatic events under changing climate conditions[49–51].

Our results call for improved vulnerability assessment approaches in Europe and other regions of intensive agriculture. Current practices that do not consider transient dynamics could lead to a substantial underestimation of the extent of vulnerable areas and the associated risk. Our study addresses this limitation and thus provides crucial vulnerability criteria, which can then be combined with information on the exposure (i.e., available data on excess nitrate components) for a risk assessment. To this end we provide a showcase example of vulnerability assessment combining information on excess nitrate (see "Methods"). We, therefore, urge modelers and planners to further develop and evaluate tools that use transient dynamics to assess the vulnerability of the subsurface to diffuse pollution (e.g., excessive nutrient surpluses in the root zone). Further avenues of research include the improvement, validation, and quantification of the predictive uncertainty of this kind of vulnerability assessment. Improved vulnerability maps are fundamental for deciding on agricultural subsidies and nitrate management, which are key components of the EU's common agricultural policy (CAP).

## Methods
**Continental-scale hydrologic simulations**. We use the spatially explicit, process-based, mesoscale hydrologic model (mHM[52,53]) to perform the continental-scale hydrologic simulations over Europe. The model features the multiscale parameter regionalization (MPR) technique, which explicitly accounts for the spatial heterogeneity of fine-scale terrain, soil, vegetation, and other landscape properties. The mHM-MPR modeling framework provides a unique capability to simultaneously and seamlessly operate the model at multiple scales and locations[52–54]. We briefly describe the processes within the root-zone soil compartment, relevant for this study, in the following text. Here, we account for the dynamics of the first meter of soil that coincides with the rooting zone for arable lands. This compartment is modeled as three consecutive layers with the following depths: 0–5, 6–25, and 26–100 cm. In each layer, the incoming water in the form of rainfall plus snowmelt, after accounting for the canopy interception, is partitioned into soil-water storage and exfiltration based on a non-linear (power) function depending on the degree of

saturation of the corresponding layer, following the conceptualizations used in other large-scale models[55,56]. The exfiltrated water from the first layer is input to the second layer and from the second to the third layer. The evapotranspiration losses from each layer are modeled as a fraction of potential evapotranspiration and are dependent on water storage deficit-induced stress and the fraction of the vegetation roots in each layer (see Supplementary Fig. 10 for a conceptualization of these root-zone soil moisture accounting processes). Readers interested in a complete model description may refer to previous studies[52,53,57], and www.ufz.de/mhm for the source code and a detailed user manual. A comprehensive overview of underlying datasets, processing steps, as well as model establishment including impact assessment of natural/human intervention activities (e.g., irrigation) is detailed in Supplementary Note 1 and in Supplementary Table 1. Model simulations are produced at a 0.25° spatial resolution and on a daily timescale for the period 1950–2015, using the model parameterizations established in previous studies[53,54,58] (see also the Supplementary Note 1 for more details). We conducted a thorough multivariate evaluation of model performance across the European domain (see Supplementary Note 2 for details). While the choice of spatial resolution used here is constrained by the availability of meteorological forcing datasets (E-OBS; v13.0)[59], the multiscale parameterization approach implemented in mHM allows for the explicit treatment of fine-scale, sub-grid variability of landscape features[52,53]. We emphasize that our study focuses on providing a general framework for the characterization of subsurface nitrate vulnerability and applied it to the pan-European landscape. This framework can also be applied/expanded to finer scales (e.g., catchment scale), incorporating more detailed datasets[27] and relevant processes to provide crucial insights into local vulnerability and assist policy intervention strategies. Examples are the localized effects of irrigation (see Supplementary Note 1 for details) and artificial (tile) drainage, which can impact soil-water storage and indirectly the root-zone transport dynamics.

Hydrologic components of mHM has been also coupled to a nitrate transport model and previous studies have demonstrated successful applications of this coupled model[43,60]. The formulation of reactive timescales related to denitrification process in soil used in this study follows a similar approach to that of the coupled model (i.e., a first-order denitrification in soil along with a space-time variability of environmental factors; see the Section below on "Nitrate vulnerability assessment" for more details).

**Derivation of hydrologic transport times**. We follow recent theoretical developments to infer the time-variant nature of the travel-time distributions (TTDs)[21,25,26,61–63]. Specifically, we adopt the notion from Botter et al.,[21,61] who provided an elegant expression for deriving the time-variant TTDs for water parcels entering or leaving a control volume based on the temporal evolution of water storages and fluxes under several mixing (age function) schemes. We use the transient formulations of TTDs on a control volume taken as a single-grid cell and soil layer, which is characterized by the daily dynamics of the soil-water storage ($S$) and incoming flux as effective precipitation $J$ (snowmelt plus rainfall minus canopy interception losses) and outgoing flux as $O = (I + E)$, where $I$ and $E$ represent the exfiltration and evapotranspiration fluxes from a given soil layer, respectively. Under a random sampling scheme of mixing that assigns the same probability to all water particles with different ages in storage to be sampled by outgoing fluxes, the analytical expression for the transient TTD at any time $t$ for water parcels exiting (as the exfiltration flux) can be expressed as follows[21]:

$$p_I(t - t_{in} | t_{in}) = \frac{I(t)}{\theta(t_{in})S(t)} \exp\left(-\int_{t_{in}}^{t} \frac{I(t') + E(t')}{S(t')} dt'\right) \quad (1)$$

with $t - t_{in}$ ($t > t_{in}$) represents the time from the moment the water parcel enters the control volume ($t_{in}$) until now ($t$). The partition function $\theta(t_{in})$ indicates the portion of the water parcel that enters the control volume at $t_{in}$ and leaves as the exfiltration flux (specifically, $I$ as opposed to $E$) and is expressed as follows:

$$\theta(t_{in}) = \int_{t_{in}}^{\infty} \frac{I(t)}{S(t)} \exp\left(-\int_{t_{in}}^{t} \frac{I(t') + E(t')}{S(t')} dt'\right) dt. \quad (2)$$

This partition function $\theta$ defines a dimensionless number between [0 and 1]. The above expression is formulated for the TTDs conditioned to the entrance (or injection) time of water particles to the control volume, and therefore, it relates to the concept of life expectancy that tracks the ages of water particles forward in time[64,65]. The complementary approach in which the ages of the water particles exiting the system are tracked back in time relates to an age concept[65], and both forward and backward TTs can be related through the Niemi's continuity equation[66]. Recent studies[22,23,25,63,67,68] demonstrated the usefulness of the transient TTDs in capturing the overall behavior of hydrological and geochemical responses in experimental and intensively monitored catchments.

We numerically solve the expressions to derive the daily evolution of the Europe-wide TTDs based on the water fluxes and storages of the three soil layers simulated by mHM for the period 1985–2015. The procedure to derive the overall TTD representing the entire root zone (0–1 m) is carried out in two steps. First, we derive the time-varying TTDs for each soil layer separately using the layer-specific storages and outgoing water fluxes at every modeling time-step. The overall TTD of water parcels leaving the entire root zone at a 1 m depth is then estimated by

sequential convolution of the independently estimated probability density functions for the first, second, and third soil layers. Since this procedure is followed for each grid cell and the modeling timestep separately, it leads to a very high computational effort at a continental scale and daily time steps. We then summarize the daily TTDs for each grid cell with statistical measures corresponding to the median ($TT_{50}$) and interquartile range ($TT_{IQR}$), as well as the tails of the distribution such as the 10th ($TT_{10}$) and 90th ($TT_{90}$) percentile estimates.

One of the critical considerations of the above TTD formulation is the choice of the mixing schemes (or StorAge Selection; SAS functions[25,61,62,69,70]) for the preference of water parcels with different ages being sampled by outflows. Among several mixing approaches, we consider here a random sampling scheme for deriving TTDs in each soil layer meaning that all water parcels in storage have an equal preference for sampling; consequently, all the following analyses of the TTD characterization are contingent on this selection. Our decision is motivated by the fact that there is no a-priori information available for the mixing schemes (or SAS functions) at a continental scale and that the random sampling scheme is one with the highest entropy. It is important to note that despite the random sampling scheme used here for characterizing the TTDs in individual soil layers, the overall sampling scheme for the entire soil column is far from being random due to difference in soil-water content and evapotranspiration fluxes in different soil compartments[26]; and such an approach provides a meaningful way to simulate non-random sample dynamics as shown in a recent study[70].

**Nitrate vulnerability assessment**. Our nitrate vulnerability assessment is based on the leaching of excess nitrate from the root zone after accounting for the plant uptake (and other turnover processes). The mechanisms considered are downward transport and removal by denitrification, and we contrast the corresponding timescales of hydrologic transport (TT) with denitrification reaction (RT). Owing to the lack of reliable observations of RT, especially at large scales, we take a scenario approach and consider a wide range of RT estimates varying between 0.5 and 5 years. We represent the RT as a characteristic reaction timescale[38] corresponding to a given percentage of removal of initial substrate—here taken as 50% ($RT_{50}$) to allow for comparability with the corresponding hydrologic transport timescale ($TT_{50}$). Following the first-order kinetics adopted in this study, a $RT_{50}$ of 1 year, for example, would correspond to a denitrification rate constant of $-\ln(0.5)/RT_{50} = 0.69 \text{ y}^{-1}$. Following previous large-scale studies[39,42,46], RT represents an effective timescale $\langle RT_{50} \rangle$ that encapsulates the relevant environmental factors, such as soil moisture, temperature, and organic carbon content, that affect the site-specific reaction behaviors[48].

We consider two cases for the nitrate vulnerability assessment that account for the static and transient behaviors of transport and reaction timescales. In the static case, we use the average estimates of the transport times $\mu(TT_{50})$ and contrast them with the effective $\langle RT_{50} \rangle$ values. In the transient case, we account for the daily dynamics of $TT_{50}(t)$ and $RT_{50}(t)$ for each grid cell. The transient nature of the site-specific $RT_{50}(t)$ is constructed from for the spatiotemporal variability of the environmental factors, $f_e(t)$, that affect the daily variability of the denitrification process. Specifically, we account for the dynamic reduction factors caused by varying soil moisture $f_S(t)$ and temperature $f_T(t)$ conditions. Following established parameterization approaches[39–41,43,48], we define the following time-varying, dimensionless power functions $f_S(t)$ and $f_T(t)$ varying between [0,1] to reflect the status of anoxic and optimal temperature conditions required for the denitrification process[48]:

$$f_S(t) = \begin{cases} 0 & S(t) < S_\tau \\ \left(\frac{S(t) - S_\tau}{S_m - S_\tau}\right)^\omega & S_\tau \le S(t) \le S_m \\ 1 & \text{otherwise,} \end{cases} \quad (3)$$

and,

$$f_T(t) = \beta^{\frac{T(t) - T_r}{10}} \quad (4)$$

where $S(t)$ is the soil-water storage at a given day, $t$, relative to the saturation limit ($S_m$), and $S_\tau$ is the threshold limit below which the denitrification process is completely inhibited, taken here as one-third of the saturation limit[41,48]. The parameter $\omega$ (= 2.5) defines the steepness of the curve[41,43,48]. $f_T(t)$ represents the effect of increasing ambient temperature $T(t)$ on denitrification. $T_r$ (= 25 °C) represents the reference temperature where $f_T(t) = 1$, and $\beta$ (= 2) is the factor that embeds the dependence of the denitrification rate on ambient temperature[39,40,48]. We also consider the effect of spatially varying soil organic carbon content by accounting for the spatial information of potential denitrification rate constants based on the agro-ecosystem Carbon And Nitrogen DYnamics (CANDY) model[71]. Across the majority of the cultivated areas in Europe, the organic carbon content varies between 0.5 and 8%, with a median estimate of ~1% (see Supplementary Fig. 11) and the corresponding rates range between 0.02 and 0.16 kg/ha/day for the 1 dm soil layer. We use the relative (spatial) variations in this rate information to derive the spatial multiplier factors ($f_{OC}$) and we reference these factors to the nominal value of 1.0 for the base carbon content of 0%. Following an established approach[41,43,48], we estimate the combined effect of multiple environmental factors for each grid cell and modeling timestep as $f_E(t) = f_S(t)f_T(t)f_{OC}$. However, this $f_E(t)$

value only describes the environmental condition on a single day $t$, whereas any solute entering on a given day $t_{in}$ will reside in the root zone for a much longer time period and be, therefore, exposed to a range of environmental conditions. We express this combined effect of the varying environmental conditions over the entire transport time as:

$$\langle f_E \rangle(t_{in}) = \int_{t_{in}}^{\infty} p_I(t - t_{in}|t_{in}) \left( \frac{1}{t - t_{in}} \int_{t_{in}}^{t} f_E(t')dt' \right) dt. \quad (5)$$

The environmental conditions $f_E(t)$ in the above formulation are, therefore, weighed such that their different impact over the entire transport time is acknowledged. By virtue of the inner integral in the above equation, water, and therefore solutes (excess nitrate), leaving at any given day is only affected by the averaged behavior of the environmental condition up to that day (i.e., from $t_{in}$ to $t$). The combined behavior is then represented as an aggregated behavior of all the different portions of water parcels or solutes leaving the root zone; and this effect is reflected in the outer integral in the above equation. We use as grid-specific environmental factor the normalized, time-varying weighting factor $\langle f'_E \rangle(t)$, to establish the site-specific temporal dynamics of the denitrification timescale $RT_{50}(t)$. The normalization is based on the site-specific averaged $\langle f'_E \rangle(t)$ values that allows for the preservation of the effective $\langle RT_{50} \rangle$ across the study domain.

Our subsurface nitrate vulnerability analysis makes use of the dimensionless Damköhler number ($D_a = \frac{TT_{50}}{RT_{50}}$) to depict the complex interplay between the hydrologic transport (leaching) and biogeochemical turnover (denitrification) timescales[34–36,38]. This dimensionless number ranges between 0 and $\infty$ with $D_a < 1$ indicate the dominance of the hydrologic transport over the reaction timescales, and vice versa. We use this objective measure to delineate the regions across Europe (with $D_a \leq 1$) that are vulnerable to nitrate leaching from soil. We synthesize this information for the cultivated area using the cropland map[72] of the year 2000, which was compiled using extensive resources at national and sub-national level agricultural census statistics as well satellite-based land cover classification datasets[72]. We consider as cultivated grid cells only those that have a cropland fraction of at-least 5%. Based on this threshold, a total number of nearly 8100 cropland cells at 0.25° grid resolution (leading to a total of ~192 million ha of cultivated areas) are considered in our vulnerability assessment analysis.

**Vulnerability assessment contingent on N-surplus.** Our analysis mainly considers the nitrate vulnerability across all cultivated areas without accounting for the spatial heterogeneity of N-surplus (or excess N) estimates, i.e., net nitrogen balance after accounting for fertilizer and atmospheric inputs and plant uptake. Although these estimates are needed to properly characterize the risk of nitrate leaching, they are rarely available at sub-annual timescales. Taking the example of the N-surplus balances for croplands around the year 2000[73], we note that not all of the cultivated areas across the study domain have high N-surplus, especially the east European regions bordering the Black sea (see Supplementary Fig. 12 for the N-surplus map and the corresponding cropland fractions in Fig. 3b). There is, however, a general tendency of increased N-surplus with the larger fraction of cropland areas (see Supplementary Fig. 12). We find that around 33% of the total cultivated areas have a high N surplus ($\geq 1 \times 10^5$ kg-N per grid cell) and is vulnerable to nitrate leaching under the static vulnerability assessment case (i.e., cropland cells of $D_a \leq 1$ based on the averaged $TT_{50}$ and effective $\langle RT_{50} \rangle$ value). For the transient case based on the time-varying $TT_{50}$ and $RT_{50}$, we find around 58% of the total cultivated areas with high N-surplus is temporarily vulnerable ($D_a(t) \leq 1$) to nitrate leaching for at least one-third of the year. The analysis conducted here provides a preliminary overview on the nitrate risk assessment, and further improvements in this aspect require more information on the temporal variability of N-surplus.

## Data availability
Data used in this study have been obtained from the following open sources: Terrain elevation GOTOPO30 and EU-DEM from https://www.usgs.gov/ (https://doi.org/10.5066/f7df6pqs) and http://www.eea.europa.eu/data-and-maps/data/eu-dem; the river database CCM2 v2.1 from https://ccm.jrc.ec.europa.eu; Soils texture maps based in HWSD from https://webarchive.iiasa.ac.at/Research/LUC/External-World-soil-database/HTML/; the land cover product GlobCOVER v2 from http://due.esrin.esa.int/page_globcover.php; the CORINE land cover products (v18.4) from http://land.copernicus.eu/pan-european/corine-land-cover; the hydrogeology map IHME1500 v11 from http://www.bgr.bund.de/ihme1500; the historical forcings E-OBS v13 from https://www.ecad.eu/E-OBS/; and the 2000 cropland and N-excess maps from http://www.earthstat.org. Further details on these and other supporting databases are provided in Supplementary Table 1. Finally, the underlying data for drawing the main conclusions of this study are available at http://dx.doi.org/10.5281/zenodo.3531907; and other auxiliary model simulations can be available from the corresponding author upon request.

## Code availability
The underlying model source code along with a test case to validate successful installation of the mesocale Hydrologic Model (mHM) and the detail user manual are available at https://git.ufz.de/mhm/. The source code for computing the travel-time distributions (TTDs) is adapted after Hesse et al.[27], which is available at https://git.ufz.de/

mhm/mhm/tree/develop/post-proc/sas/. Further support on the details of processing algorithms can be obtained from the corresponding author upon request.

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

## Acknowledgements

R.K. acknowledge the support from the Initiative and Networking Fund of the Helmholtz Association through the project Advanced Earth System Modeling Capacity (ESM) (www.esm-project.net). P.S.C.R. acknowledges partial financial support from the Lee A. Reith Endowment in the Lyles School of Civil Engineering, Purdue University. Collaborations between the Helmholtz Center for Environmental Research-UFZ, the Purdue University, and the University of Florida are supported by an international Memorandum of Understanding (MOU). Support to F.S. was provided by the Reduced Complexity Models project co-funded by the Helmholtz Association. We also acknowledge the partial funding for collection and processing of datasets under a contract for the Copernicus Climate Change Service (C3S 441 Lot1 NERC; http://edge.climate.copernicus.eu). ECMWF implements this service and the Copernicus Atmosphere Monitoring Service on behalf of the European Commission. We would like to thank people from various organizations and projects for kindly providing us with the data that

were used in this study, which includes among others: USGS, ESA, JRC, NASA, GRDC, BGR, EARTHSTAT, and IIASA. We acknowledge the E-OBS dataset from the EU-FP6 project ENSEMBLES (http://ensembles-eu.metoffice.com) and the data providers in the ECA&D project (http://www.ecad.eu). Data analysis was conducted at the High-Performance Computing (HPC) Cluster EVE, a joint effort of both the Helmholtz Center for Environmental Research-UFZ and the German Center for Integrative Biodiversity Research (iDiv) Halle-Jena-Leipzig.

## Author contributions

R.K. conceived the idea and designed the study with inputs from F.H., P.S.C.R., A.M., J.W.J., and F.S. R.K. collected required datasets, conducted model simulations, and performed the analysis. R.K. led the writing with F.H. and P.S.C.R. F.H., P.S.C.R., A.M., J.W.J., F.S., L.S., J.H.F., O.R., S.T., and S.A. contributed to the editing and commenting on the manuscript.

## Funding

## Competing interests

The authors declare no competing interests.
