## [Peer Review File · Nature Communications]

Reviewers' comments

Reviewer #1 (Remarks to the Author):

The authors present a continental-scale model for the transport of water through the shallower part of the soil (root zone). The model has a relatively high spatial resolution of (0.25°) and it is built upon a hydrologic model previously established by the same authors. The model is used to compute water residence time distributions through the root zone across the entire Europe. Then, the authors compare the median transit time to the typical timescales that characterise denitrification to identify the areas that are most vulnerable to excess nitrate leaching. Interestingly, the authors find that the vulnerability estimated through steady-state simulations (which is typically done in the practice) largely underestimates the vulnerability estimated through transient simulations, with notable implications for the implementation of mitigation strategies.

Among the key results of this paper I identify: 1-the large spatial scale and resolution of the model, which was never achieved before at the root-zone level; 2-the resulting opportunity to investigate and compare residence time distributions across the entire Europe. I believe this work addresses one important knowledge gap in hydrologic transport: the lack of comparability of results among different locations. I am also sure that results from this model will lead to additional important insights into the transport of water and solutes in the landscape; 3-the new framework to systematically evaluate the vulnerability to nitrate leaching over space but also time.

These results are certainly of immediate interest in the fields of hydrology, water quality and geochemistry. The approach is valid and I did not identify any major flaw that may invalidate the author's conclusions. The reproducibility of the results needs to go through the tremendous amount of work done by the authors to harmonize the different data sources. But datasets are publicly available and the model is openly accessible and well-documented. Figures are appropriate, clear and nicely done. I am supportive of publication of this work and I suggest improvements below.

SPECIFIC COMMENTS

MODEL REALISM and SCALE

I invite the authors to better stress through the paper that these are the re-

sults of a model (not recalling that results come from a model is a widespread mistake in hydrologic transport research). Examples (not exhaustive): stating that TTDs are "observed" is misleading (line 86); what is indicated as travel times "at" a certain region (line 98), could rather be the travel times "modelled/estimated at" a certain region; line 150: "...the overall transport behaviour observed across Europe.", again please use "simulated" rather than "observed".

As for many modelling studies, there is the obvious criticism that results are unverifiable. Thus I think the authors should highlight previous successful applications of the transport model, such as their reference [46]. Then, there are two (non-central) results that, in my view, might potentially be related to the model implementation rather than from actual transport processes. These are: 1) the coefficient of variation (CV) of the median travel time, which is approximately 0.4 across the entire Europe; 2) the very strong linear correlation among the mean median travel time and the aridity index. I'd appreciate if the authors could briefly discuss whether these results could be a model artefact.

I expect the authors to further comment on the following 2 aspects:

1-how the median transit time estimates should be interpreted, considering that grid cells can be as large as 500 km² and certainly there is a large subgrid variability due to the variety of subsurface properties and land uses within a single cell? This subgrid variability could be indeed crucial for the estimation of local vulnerability to nitrate inputs.

2-the lack of modelling of irrigation seems important given that the focus of the work is on cultivated areas. So invite the authors to expand on what is the expected effect of irrigation and what could be done to mitigate this missing component in the model.

Da(t) DEFINITION AND ANALYSIS

I appreciate the effort in combining complex spatio-temporal dynamics into a single indicator like the Damköhler number, that can be easily visualized. I just wonder whether this is the most appropriate indicator for transient dynamics. Is it meaningful to compute a different Da for every day of simulation, when Da represents the ratio between timescales of tens or hundreds of days? Also, the median transit time associated with a day of precipitation is related to the reaction timescale for the same day. But then this does not take into account that the environmental conditions that favour/limit nitrate removal change throughout the transport process rather than remaining fixed to the value when rain fell. Or is this just my incorrect interpretation?

FIGURE 3

This is a very nice and very important figure for the manuscript, but it includes a large amount of information and I have a few suggestions to make it more readable. I feel figures 3c and 3d are probably not needed in the main text and could be moved to the supplement. Figure 3d is not very intuitive to read (the legend is slightly different from what is plotted) and maybe the smooth statistics are not needed. Also, I find it difficult to visually compare figures 3f with figure 3f and my suggestion (that the authors should feel free not to follow) is to add a map that somehow compares the difference between the static and the transient case. This would make it easier to localize the effect of time-variance and would better support lines 223-226. Finally, consider removing from the maps the cells with < 5 cropland fraction.

REFERENCES

-line 51: the recent review paper by Sprenger et al. (2019) might be a good

addition here.

-line 53: maybe you do not need 11 citations here (note that some have a strange formatting)

-line 36: I find it unclear whether the references here come from the previously cited paper or not

-line 58: please mention work by the group of Reed Maxwell as they developed a transient transport model at the continental scale (although not focusing on reactive transport).

-line 342: you can refer to the work by Remondi et al. because they used a similar approach

MINOR REMARKS AND TYPOS

- I found a bit unclear how the root zone is defined in the model. Is it simply the first meter of soil (which is a good approximation for arable lands, i.e. for the focus of the study)? If so, you could say that you model the dynamics of the first meter of soil and that for arable lands this coincides with the rooting zone depth. Then I'm just curious whether the model, in its current implementation, would be able to cope with a spatially-variable rooting zone depth.

-171: include "here defined as" within the brackets before the definition of D_a .

-176 based on A previous meta-analysis (may be worth expanding this a bit)

-194-195: I guess you refer to Figure S3 from the cited EU report. I find it difficult to compare this map with the one provided in the manuscript (which is said to be "remarkably consistent") and I think that if you want to compare them, then they should be made more comparable (i.e. similar color code).

-256: "Our results call for a paradigm shift in vulnerability assessments": is this really a paradigm shift? Farmers have been knowing for long time that there are periods (typically wet winters) when fertilizations should not take place.

-351-353: this sentence sounds a bit vague and unjustified, so I suspect it would be better to remove it.

-356-358: this is not just "worth mentioning". This is very important and was previously shown (e.g. Remondi et al. 2018) to be a meaningful approach to simulate very non-randomly-sample dynamics.

-line 94: ranges (..) increases.

-250-252 there seems to be something wrong with this sentence

-equation (2): it looks similar to eq (1) so I guess there should be a minus within the exponential argument. And it looks like the variable theta is just a normalization factor in the end, so the formula might not be needed.

-323: complementary

-the environmental factors are sometimes indicated as f_e or as fE . Similarly, both f_s and fS are used.

BIBLIOGRAPHY

-Sprenger M, Stumpp C, Weiler M, Aeschbach W, Allen ST, et al (2019) The demographics of water: A review of water ages in the critical zone. *Rev Geophys* 2018RG000633. doi: 10.1029/2018RG000633

-Remondi F, Kirchner JW, Burlando P, Fatichi S (2018) Water Flux Tracking With a Distributed Hydrological Model to Quantify Controls on the Spatiotemporal Variability of Transit Time Distributions. *Water Resour Res* 54:3081-3099. doi: 10.1002/2017wr021689

Reviewer #2 (Remarks to the Author):

In this paper, the authors use estimates of static and transient root-zone hydrologic travel times to identify the locations of nitrate-vulnerable zones across Europe. First, the authors show that humid areas with higher rainfall have shorter mean travel times. Conversely, semi-arid areas have much longer travel times and also a greater standard deviation for the daily soil-water travel times. They also show seasonal variability in travel times, with dryer summers demonstrating longer travel times. In other words, the authors show that when there is more precipitation, there is faster movement of water and solutes to the subsurface. With regard to denitrification dynamics, the authors note that denitrification rates are poorly constrained, especially at large scales. Accordingly, they do not use "real" estimates of denitrification rates, but instead take what might be considered a "toy model" approach, and consider a range of reaction time scales, i.e. how long will it take to denitrify 50% of the initial nitrate substrate? Using the relationship between the travel time and the reaction time scale, as represented by the Damköhler number, Da , the authors explore how nitrate transport from the root zone to groundwater might vary across the European landscape. Based on their coupled hydrologic-biogeochemical metric, the authors speculate that areas with faster travel times will be more vulnerable to groundwater nitrate contamination. The issue of nitrate contamination of groundwater is an important one. However, it is unclear whether the current manuscript represents a significant advance over current work or is reliable in its final conclusions.

Specific Comments:

1) The authors provide estimates of static and transient hydrologic travel times within the root zone based on model results using mHm, a meso-scale hydrologic model. The variability of these travel times across Europe is interesting and provides a useful basis for understanding solute transport across the continent. It is concerning, however, that the model only simulates natural hydrologic behaviour and does not account for major human intervention activities. Areas at the highest risk for nitrate leakage to groundwater (cropland) are also the most likely to be artificially drained. Areas with artificial subsurface drainage systems will have drastic decreases in travel times, but will also have preferential routing of high-nitrate runoff to surface water. In this regard, the modeled travel times, and the inferred risk to groundwater, will in many cases be completely unrelated to actual vulnerability. Significant amounts of cropland across Europe are artificially drained, so this lack of attention to tile-drainage effects is a significant lack in the current study. As noted in the manuscript, the model also does not take into account increased soil water content due to irrigation, which would clearly affect any quantification of travel times. Though this lack is acknowledged by the authors, such acknowledgement does not add confidence to our interpretation of the current reported results.

2) The authors provide a map of groundwater vulnerability that is produced by coupling the calculated hydrologic travel times with a generic, moderate estimate of the denitrification timescale. The authors suggest, in the introduction, that their work can provide an integrated understanding of the complex and dynamic interplay between hydrologic transport and biogeochemical turnover. They also indicate that their RT50 metric, which is used to characterize the denitrification timescale, is an effective timescale encapsulating relevant environmental factors such as soil moisture, temperature and organic content, that affect the observed site-specific reaction cases. In reality, however, there is no site-specific estimate of denitrification. The mapped vulnerabilities to nitrate leaching provided in the paper assume an average denitrification rate constant that is uniformly applied across Europe, regardless of variations in soil moisture, temperature, and soil organic matter content. As a result, the findings of the paper are essentially based solely on the differing travel times rather than any real consideration of spatially varying biogeochemical turnover.

3) The authors appear to assume that any cultivated area has uniformly available excess nitrate that may leach to groundwater under appropriate hydrologic and biogeochemical conditions. This assumption does not take into account variations in source nitrate across different crop types and cropping intensities. Nitrate availability must be as important as transport pathways and travel times when considering the vulnerability of groundwater to nitrate contamination, so this lack of consideration of sources is a significant weakness of the paper.

4) There is no attempt at validation of the current results. First, the mHM model is only a hydrologic model and does not include biogeochemical processes. As a result, there is no explicit modeling of stream nitrate or partitioning of nitrate fluxes between surface and subsurface pathways. An alternative approach, of course, would be to validate the toy-model predictions with actual maps of groundwater nitrate concentrations. To strengthen the current findings, I would recommend comparisons of predicted high-vulnerability areas with measured nitrate concentration values. Finally, Ascott et al. published a 2017 paper on "Global patterns of nitrate storage in the vadose zone," in Nature Communications. While the current paper does cite the Ascott paper, it provides no discussion of how the current results correspond to the Ascott estimates of nitrate accumulation in the vadose zone. This lack of mention suggests a lack of attention to current relevant literature and also a loss of opportunity for outside validation of the current results.

Reviewers' comments are listed in black followed by our response in blue.

Reviewer #1 (Remarks to the Author):

The authors present a continental-scale model for the transport of water through the shallower part of the soil (root zone). The model has a relatively high spatial resolution of (0.25°) and it is built upon a hydrologic model previously established by the same authors. The model is used to compute water residence time distributions through the root zone across the entire Europe. Then, the authors compare the median transit time to the typical timescales that characterise denitrification to identify the areas that are most vulnerable to excess nitrate leaching. Interestingly, the authors find that the vulnerability estimated through steady-state simulations (which is typically done in the practice) largely underestimates the vulnerability estimated through transient simulations, with notable implications for the implementation of mitigation strategies.

Among the key results of this paper I identify: 1-the large spatial scale and resolution of the model, which was never achieved before at the root-zone level; 2-the resulting opportunity to investigate and compare residence time distributions across the entire Europe. I believe this work addresses one important knowledge gap in hydrologic transport: the lack of comparability of results among different locations. I am also sure that results from this model will lead to additional important insights into the transport of water and solutes in the landscape; 3-the new framework to systematically evaluate the vulnerability to nitrate leaching over space but also time.

These results are certainly of immediate interest in the fields of hydrology, water quality and geochemistry. The approach is valid and I did not identify any major flaw that may invalidate the author's conclusions. The reproducibility of the results needs to go through the tremendous amount of work done by the authors to harmonize the different data sources. But datasets are publicly available and the model is openly accessible and well-documented. Figures are appropriate, clear and nicely done. I am supportive of publication of this work and I suggest improvements below.

Response: We are thankful to the Reviewer for his/her positive assessment and support for publication of our work. We have carefully considered each of the Reviewer's comment, and revised our paper accordingly. We hope that we have addressed all the comments satisfactorily, as detailed below.

SPECIFIC COMMENTS

MODEL REALISM and SCALE

I invite the authors to better stress through the paper that these are the re-

sults of a model (not recalling that results come from a model is a widespread mistake in hydrologic transport research). Examples (not exhaustive): stating that TTDs are “observed” is misleading (line 86); what is indicated as travel times “at” a certain region (line 98), could rather be the travel times “modelled/estimated at” a certain region; line 150: “...the overall transport behaviour observed across Europe.”, again please use “simulated” rather than “observed”.

Response: We agree with the Reviewer’s point, and we have made sure the reviewer’s suggestion is incorporated in the revised document to make this point clear. Our (past) intention by using wordings like “observed” was from the textual point of view wherein we want to inform about the results of the model based analysis.

As for many modelling studies, there is the obvious criticism that results are unverifiable. Thus I think the authors should highlight previous successful applications of the transport model, such as their reference [46]. Then, there are two (non-central) results that, in my view, might potentially be related to the model implementation rather than from actual transport processes. These are: 1) the coefficient of variation (CV) of the median travel time, which is approximately 0.4 across the entire Europe; 2) the very strong linear correlation among the mean median travel time and the aridity index. I’d appreciate if the authors could briefly discuss whether these results could be a model artefact.

Response: We truly appreciate and value this kind feedback. Regarding the model evaluation, we have made every possible effort in evaluating the model predictions against (hydrologic) measurements available to us (as elaborated in detail in the Supplementary materials). As suggested by the Reviewer, we highlight the previous successful applications of the transport model, coupled to mHM, (Yang et al. 2018, 2019), which supports the validity of our analysis results. In this regard we specified “Hydrologic components of mHM has been also coupled to a nitrate transport model and previous studies have demonstrated successful applications of this coupled model (Yang et al. 2018, 2019). The formulation of reactive timescales related to denitrification process in soil used in this study follows a similar approach to that of the coupled model (i.e., a first-order denitrification in soil along with a space-time variability of environmental factors).”

Regarding the two other queries that are non-central to the overall message delivered in this research work – as the reviewer also rightly pointed out – we recognize that our findings are based on a well-established modeling chain for continental-scale travel time analysis; and in this regard there are uncertainties related to e.g., input geomorphological and forcing datasets, hydrologic model and transport timescale parameterizations, among other things. Since these are crucial to any model-based assessment, we discussed them in detail in the re-

vised manuscript (see the Methods as well the supplementary sections on Data processing/Model establishment and evaluation).

To elaborate further on the first question, our continental scale analysis indicated a high correspondence between mean and standard deviation of the median travel times (and therefore a consistent estimate of resulting coefficient of variation). We do not think that the consistency of the coefficient of variation (CV) noticed across the (large) study domain is model dependent in the sense that another hydrological model would result in similar consistency across such a large scale. However, the value and the consistency of the coefficient of variation (CV) is, at least in part, the result of its mathematical properties and may be considered model dependent in that sense. Here, the latter notion is based on the reasoning that with a higher mean values of a quantity, there is more range for this quantity to vary and consequently a higher potential variance. Such kind of consistency has been also noticed in a previous study by Botter et al., 2012 for the CV values of flow travel times across different hydroclimatic settings with experimental model simulation results. We also wish to emphasize here that our main aim here was not to provide a single estimate (of CV), but to understand and reveal possible large-scale drivers that can best explain such a consistent behavior of travel times across Europe. Regarding the second question, the strong linear relationship between median travel time and the aridity index can be motivated by physical reasoning and should therefore be considered to reflect reality instead of being a model artifact. This can be explained through physical considerations such that if there is little to no rain, nothing is pushing the water out of the soil and it will remain there for longer. In addition, given that this is a large-scale study, the impact of small-scale factors may be dampened, which will automatically increase the role of the large-scale, climatic factors encapsulated in the aridity index. The large variability in the hydroclimatic conditions across the study domain therefore act as (a first-order) dominant control that mostly explain the spatial variability of the resulting travel times. Combined, these factors conspire in making the climate, encapsulated in the aridity index, the dominant factor and give us confidence that the results reported here reflect the real-world conditions instead of being a modelling artifact. Furthermore we also note that our findings on the strong association between travel times and aridity at the continental scale can be backed up by a theoretical finding reported in past studies for example by Harman et al. (2011). As these aspects are non-central to overall message of this work – as the reviewer has also pointed out – therefore we have only briefly touched upon them in the revised manuscript.

Harman, et al. (2011):, Climate, soil, and vegetation controls on the temporal variability of vadose zone transport, *Water Resour. Res.*, 47, W00J13, 10.1029/2010WR010194.

Yang, et al. (2018): A new fully distributed model of nitrate transport and

removal at catchment scale. *Water Resour. Res.*, 54, 10.1029/2017WR022380
Yang, et al., (2019): Sensitivity analysis of fully distributed parameterization reveals insights into heterogeneous catchment responses for water quality modeling. *Water Resour. Res.*, 55, 10.1029/2019WR025575
Botter, G. (2012): Catchment mixing processes and travel time distributions, *Water Resour. Res.*, 48, W05545, doi:10.1029/2011WR011160

I expect the authors to further comment on the following 2 aspects:

1-how the median transit time estimates should be interpreted, considering that grid cells can be as large as 500 km² and certainly there is a large subgrid variability due to the variety of subsurface properties and land uses within a single cell? This subgrid variability could be indeed crucial for the estimation of local vulnerability to nitrate inputs.

Response: Our choice for the selection of the spatial resolution (25 km) was constrained by the availability of (meteorological) database and practical reasons considering that the analysis was conducted over the large study domain (i.e., entire Europe and neighbouring parts), involving large computational resources. As an example, analysis results for the daily travel time distributions (obtained from large-scale hydrologic simulation runs) across the study domain like Europe at a chosen spatial resolution, requires a storage capability in order of terabytes that was hosted and run on a super computing facility. While the future procurement of high resolution runs may enhance the spatial resolution – but at the expense of exceptionally high computational resources – we expect that the main finding of our study on “the important role of the transient behaviour of transport and biogeochemical transformation process in subsurface nitrate vulnerability assessment” would remain intact.

On the role of sub-grid variability, we cannot agree more with the reviewer as this has been one of the research pillar of our past studies. The modeling scheme we used in this study is equipped with a novel multiscale parameterization technique that explicitly consider the role of sub-grid variability of local subsurface properties and land uses (Samaniego et al., 2010) and allow the model to seamlessly run at multiple spatial resolutions ranging from 1 km to 100 km, as demonstrated in several previous studies (see e.g., Samaniego et al., 2010; Kumar et al., 2013; Samaniego et al., 2017; Rakovec et al., 2016, 2019).

We now briefly reflect these issues in the revised manuscript and added: “We emphasize that our study has focused on providing a general framework for the characterization of sub-surface nitrate vulnerability and applied it to the pan-European landscape. This framework can also be applied to finer scales (e.g., catchment scale) using more detailed data-sets (Hesse et al., 2017) and therefore can provide crucial insights into local vulnerability and assist policy intervention strategies. While the choice of spatial resolution used here is constrained by the

availability of meteorological forcing datasets (E-OBS; v13.0), the multiscale parameterization approach implemented in mHM allows for the explicit treatment of fine scale, sub-grid variability of landscape features (Samaniego et al., 2010; Kumar et al., 2013).”

Samaniego et al. (2010): Multiscale parameter regionalization of a grid-based hydrologic model at the mesoscale. *Water Resources Research* 46.5 (2010).

Kumar et al. (2013): Implications of distributed hydrologic model parameterization on water fluxes at multiple scales and locations. *Water Resources Research* 49.1 (2013): 360-379.

Samaniego, et al. (2017): Toward seamless hydrologic predictions across spatial scales. *Hydrology and Earth System Sciences*, 21(9), pp.4323-4346.

Rakovec et al. (2018): Multiscale and multivariate evaluation of water fluxes and states over European river basins. *Journal of Hydrometeorology*, 19(11).

Rakovec et al. (2019): Diagnostic Evaluation of Large-domain Hydrologic Models calibrated across the Contiguous United States. *Journal of Geophysical Research: Atmospheres*.

Hesse et al. (2017): Spatially distributed characterization of soil-moisture dynamics using travel-time distributions. *Hydrology and Earth System Science*, 21 (1), pp. 549–570.

2-the lack of modelling of irrigation seems important given that the focus of the work is on cultivated areas. So invite the authors to expand on what is the expected effect of irrigation and what could be done to mitigate this missing component in the model.

Response: This issue of irrigation was touched upon by both reviewers. And therefore we conducted a thorough investigation to analyse the impact of irrigation on the modeling results. In this regard, first of all we would like to point out that across the European arable landscapes, according to the European Statistics report (EUROSTAT 2016) in 2005, 10.1% of utilised agricultural area in the EU was irrigable but only 6.8% was actually irrigated; and over the time between 2005 and 2016 these have even decreased by 3.5% for the irrigable areas and irrigated areas by 6.1%. There are however regional variations and Spain (15.7%) and Italy (32.6%) have the largest shares of irrigable areas in the agricultural areas (EUROSTAT 2016). Nevertheless we also performed a controlled simulation to analyze the impact of irrigation on resulting transport behaviors. To this end, we adapt an irrigation water demand scheme of the well-established PCR-GLOBWB model (Wada et al., 2014; Sutanudjaja et al., 2018) and contrast the model simulation results of the irrigation (Irr) versus no-irrigation (No-Irr) scenarios. Our simulation results shown in Figure 1 confirm that large part of the cultivated area across Europe show minor differences in simulated hydrological fluxes (evapotranspiration ET or leached water I from the soil profile) due to irrigation activities. For example, the changes in evap-

transpiration across 95% of cropland cells is less than 50 mm/y. Only few cropland cells ($\approx 1.8\%$) with heavy deployment of irrigation in regions of Spain and east European countries exhibited changes in ET of more than 100 mm/y. Similarly the majority of cropland cells ($\approx 98\%$) show changes in I of less than 50 mm/y between Irr and No-Irr simulations runs. These simulation results of changes in ET and I between irrigation and no-irrigation runs provide plausible conditions considering that farmers will irrigate when atmospheric demand is high and as much such that plant can take up most of the (irrigated) water and relatively smaller portion will loss or drain to deeper subsurface. Next we analyze the effect of irrigation on the resulting transport dynamics and our simulation results indicated that increased water availability due to irrigation resulted in reduced transport times (Fig. 2). However, the majority of the cropland cells ($\approx 96\%$) exhibited changes in the median travel times (TT_{50}) of less than a month; and only few cropland cells ($< 1\%$) showed changes in TT_{50} of more than 2 months (Fig. 2). In terms of relative changes in TT_{50} with respect to the No-Irr scenario runs, almost 70% of the cropland cells show no changes in TT_{50} and vast majority ($\approx 95\%$) showed a minimal change ($\leq 5\%$). Notable changes ($> 10\%$) in TT_{50} due to irrigation were seen in only 0.7% of the cropland cells (Fig. 2). In conclusion, these simulation results demonstrate that irrigation activities have a very limited impact on the inferred large-scale patterns of transport dynamics conducted at the European scale. We describe these modelling results/discussions related to irrigation/no-irrigation scenarios in the Section “Data processing and model establishment” of the revised manuscript.

EUROSTAT 2016: https://ec.europa.eu/eurostat/statistics-explained/index.php/Agri-environmental_indicator_-_irrigation

Wada et al. (2014): Global modeling of withdrawal, allocation and consumptive use of surface water and groundwater resources. *Earth System Dynamics*, 5 (1), pp. 15–40.

Sutanudjaja et al. (2014): PCR-GLOBWB 2: a 5 arcmin global hydrological and water resources model. *Geoscientific Model Development*, 11 (6), pp. 2429–2453.

Figure 1: Spatial distribution of changes in average evapotranspiration (ΔET) and leached water (ΔI) from the soil profile between the model simulations of irrigation and no-irrigation scenarios. Bottom panels depict the histogram of changes in correspondingly fluxes over the cropland cells. Note that both fluxes show increase in the irrigation scenario compared to the no-irrigation scenario.

Figure 2: Spatial distribution of changes in mean hydrologic transport time scales (median travel time; ΔTT_{50}) between the model simulations of no irrigation and irrigation scenarios. Shown in the right are the relative changes in TT_{50} with respect to base estimates of the no-irrigation scenario. Bottom panels depict the corresponding histograms of changes in ΔTT_{50} over the cropland cells. Note that in this case the median travel times (TT_{50}) are reduced in case of the irrigation scenario (i.e., signs here are opposite compared to the above changes depicted in Fig. 1 for the simulated fluxes).

Da(t) DEFINITION AND ANALYSIS

I appreciate the effort in combining complex spatio-temporal dynamics into a single indicator like the Damköhler number, that can be easily visualized. I just wonder whether this is the most appropriate indicator for transient dynamics. Is it meaningful to compute a different Da for every day of simulation, when Da represents the ratio between timescales of tens or hundreds of days? Also, the median transit time associated with a day of precipitation is related to the reaction timescale for the same day. But then this does not take into account that the environmental conditions that favour/limit nitrate removal change throughout the transport process rather than remaining fixed to the value when rain fell. Or is this just my incorrect interpretation?

Response: This is a very relevant observation by the reviewer. We would like to start by clarifying that the concept of the Damköhler number (Da), as used here is very general and applies to any reactive transport system - may be in static or in dynamic conditions. The Da calculated on a specific day of simulation characterises the leaching potential of a solute that would be added into the system that day; while considering the hydroclimatic and environmental conditions over the entire journey since its inception from soil surface to its release from the root-zone. While the static case is relatively simple and can be easily estimated, the Da estimates for dynamic systems is however conceptually challenging due to the pronounced daily variability of environmental conditions and thereby the resulting transport processes (and travel times) and reaction processes. We show here the application of this innovative approach for nitrate vulnerability assessment for the first time at the continental scale, as also recognized by the reviewer. We thoroughly acknowledge the reviewer point to account for the temporal dynamics of the reaction time (or environmental conditions), so to be consistent with the travel time dynamics. Our current estimate of the effective reaction time on a particular day only takes into account the environmental conditions of that day, or, as the reviewer puts it is "remaining fixed to the value when rain fell". This approach was consequently improved by accounting for the combined contribution of the environmental conditions over the entire travel time period to derive a more representative value of the reaction time. A detailed description of the new formulation of the environmental factors was added in the 'Methods' section of the revised manuscript (sub-section 'Nitrate vulnerability assessment' and more specifically Equation (5)). As illustrated below in Figure 3 (details in the caption); the revised simulation results still support the overall conclusion of our study: the static assessment approach greatly underestimates the extent of cultivated areas that are vulnerable to sub-surface nitrate leaching across Europe. Using the revised effective reaction time accounting for environmental conditions throughout the entire transport time, we find approximately 75% of the cultivated areas across Europe being potentially vulnerable to nitrate leaching for at least one-third of the year. Our estimate of potentially vulnerable areas is nearly twice the

current estimate of the static case (42%), and therefore echoes our statement in the abstract of the manuscript that “*future vulnerability and risk assessment studies must account for the transient behavior of transport and biogeochemical transformation processes.*” A detailed illustration of these results as well the underlying methodology are described in the revised manuscript.

Figure 3: Subsurface nitrate vulnerability assessment across Europe under static and transient considerations of hydrologic transport (TT_{50}) and denitrification (RT_{50}) timescales. The interplay between these two timescales is captured by the dimensionless Damköhler number ($D_a = \frac{TT_{50}}{RT_{50}}$) with values below 1.0 representing areas vulnerable to subsurface nitrate leaching. (a) Spatial depiction of the Europe-wide D_a estimates for the static vulnerability case; overlaid upon the grid-specific cultivated area fractions of Europe. Grid cells with at least 5% of the cropland areas are considered cultivated areas in the analysis; and they are summarized in panel (b) as a fraction of grid cells (w.r.t. total cultivated cells) according to the observed aridity index ranges (ϕ). In the static case, approximately 42% of the total cultivated cells are vulnerable to subsurface nitrate leaching ($D_a \leq 1$; areas with red hatched lines). (c) Spatio-temporal analysis for the transient vulnerability case: frequency of daily $D_a(t) \leq 1$ considering the time-varying nature of hydrologic transport and biogeochemical transport (denitrification) timescales. Considering the transient nature of D_a , approximately 75% of cultivated areas of Europe become vulnerable to nitrate leaching for at least one third of the year (i.e., frequency estimates $\geq 4/12$). (d) Depiction of the summary quantiles (0.01, 0.1, ..., 99.99%) corresponding to daily $D_a(t)$ for every cultivated grid cell, arranged according to their aridity index values (ϕ). The actual values of the summary quantiles are depicted in the background (gray colors), and the LOESS (locally weighted smoothing)-derived statistics are shown in a transparent foreground (red colors). Nearly all of the cultivated grid cells across Europe show vulnerability to sub-surface nitrate leaching for at-least 5% of the time; and majority of them ($\geq 80\%$; with $\phi \leq 2$ in panel b) for at-least 25% of the time.

FIGURE 3

This is a very nice and very important figure for the manuscript, but it includes a large amount of information and I have a few suggestions to make it more readable. I feel figures 3c and 3d are probably not needed in the main text and could be moved to the supplement. Figure 3d is not very intuitive to read (the legend is slightly different from what is plotted) and maybe the smooth statistics are not needed. Also, I find it difficult to visually compare figures 3f with figure 3f and my suggestion (that the authors should feel free not to follow) is to add a map that somehow compares the difference between the static and the transient case. This would make it easier to localize the effect of time-variance and would better support lines 223-226. Finally, consider removing from the maps the cells with < 5 cropland fraction.

Response: Thank you for your suggestions. Following the reviewer's suggestion we have moved the figure 3c and 3d to the Supplements (Fig. S8) and this have reduced the information content of this figure. We have improved the content of panel 3c (earlier panel 3e) – and retained the information of smooth statistics – albeit with higher transparency so to improve the readability. So in essence the smooth statistics provide a concise picture on the transient nature of D_a with varying climate conditions (represented by aridity index ϕ) without going through cloud of points containing detail statistics of each grid cell. For example one can easily notice a general behaviour of an increasing range of the daily D_a statistic (e.g., Q95–Q5) with increasing ϕ (i.e., when moving from humid to (semi-)arid regions). We appreciate the reviewer's suggestion regarding the visualisation of the static and the transient maps. However we would like to point out that the information provided in these maps are different and are not compatible. In the static case we provide information on actual values of the Damköhler number (D_a), whereas in the transient case we show the frequency estimate i.e., number of times the daily D_a values are less than 1.0 over the simulation period and we represent this in terms of number of months in an year. Accordingly in the latter (transient) case, the frequency estimates are from only one side of the D_a values (i.e., < 1) as we focus on assessing vulnerability to the (transient) leaching conditions. The panel 3c was plotted for precisely such a case, so that one can better appreciate the differences between the static and the transient case. Here in panel 3c, a series of daily D_a statistics are plotted (on Y-axis) for every grid cell against their respective locations in the climate space (ϕ ; on X-axis). For example, the median D_a estimates, representing the static case, can be contrasted against other D_a statistics (e.g., range between 5th and 95th quantiles) to better appreciate the temporal dynamics of the daily D_a (transient case) and how those estimates vary across different climate conditions (ϕ). In the manuscript, we have included discussions on these aspects. Finally we really value the reviewer suggestion regarding omission of the cells with < 5 cropland fraction. However for the sake of completeness we have retained the entire domain map. Although here we focus around cultivated areas

across Europe, there is also substantial amount of nitrogen entering into other natural landscapes through atmospheric depositions; and therefore we believe our spatially seamless maps and approach could be relevant for vulnerability assessments in such landscapes.

REFERENCES

-line 51: the recent review paper by Sprenger et al. (2019) might be a good addition here.

Response: Thank you for bringing this to our notice. We have included it in the reference list as suggested.

-line 53: maybe you do not need 11 citations here (note that some have a strange formatting)

Response: Thank you for bringing this to our notice. We have revised the text and kept only the relevant citations.

-line 36: I find it unclear whether the references here come from the previously cited paper or not

Response: All the papers cited here have highlighted the point of excess nitrate as a major concern.

-line 58: please mention work by the group of Reed Maxwell as they developed a transient transport model at the continental scale (although not focusing on reactive transport).

Response: Thank you. We have now included a work of the Reed Maxwell's group.

-line 342: you can refer to the work by Remondi et al. because they used a similar approach

Response: Thank you for bringing this work to our notice. We have now included it in our reference list as suggested.

MINOR REMARKS AND TYPOS

- I found a bit unclear how the root zone is defined in the model. Is it simply the first meter of soil (which is a good approximation for arable lands, i.e. for the focus of the study)? If so, you could say that you model the dynamics of the first meter of soil and that for arable lands this coincides with the rooting zone depth. Then I'm just curious whether the model, in its current implementation, would be able to cope with a spatially-variable rooting zone depth.

Response: Yes, we have taken the first one meter of soil column as definition for the root zone in arable lands. As suggested by the reviewer we now clearly mention this in the revised manuscript: "... we account for the dynamics of the first meter of soil that coincides with the rooting zone for arable lands". Yes, the

model is flexible enough and provide possibility to handle the spatially-variable rooting zone depth, depending on the user-provided soil databases. The model also takes into account the spatially varying root fractions based on the overlying vegetation types and thereby have effects on the space/time dynamics of soil-moisture accounting processes (e.g., evapotranspiration, soil moisture, in-/ex-filtration). These options are implemented in the modeling chain (for more details please refer to the manual of mHM available at www.ufz.de/mhm).

-171: include “here defined as” within the brackets before the definition of Da.
Response: Thank you. We implemented the suggested change accordingly.

-176 based on A previous meta-analysis (may be worth expanding this a bit)
Response: We have expand on this issue in the revised manuscript.

-194-195: I guess you refer to Figure S3 from the cited EU report. I find it difficult to compare this map with the one provided in the manuscript (which is said to be “remarkably consistent”) and I think that if you want to compare them, then they should be made more comparable (i.e. similar color code).
Response: We have made an effort to revise the figure so that the color coding of two maps are comparable. We present below both maps - side by side - for the visual comparison (see the caption for more details). We have also included the revised figure and the details in the revised manuscript (Supplementary Fig. S7).

Figure 4: Left (a): Map showing the leaching risk potential for agricultural land within the Environmental Zones in the EU-27 provided in a report by the European Commission (EC 2011). Right (b): Spatial variability of the Damköhler number (D_a) depicting the soil nitrate leaching potential map across Europe, computed in our study. The lower D_a values represent the areas vulnerable to subsurface nitrate leaching and vice-versa (see the main manuscript for more details). We see a high similarity between both maps especially in the Central and Western European regions, as well in Mediterranean ones. There are also differences between both maps mostly in eastern European regions. However reconciling these differences are not possible due to the lack of underlying data-sets and methodology used in creating the EC map. These regions could be focus of more targeted future studies using local datasets.

EC (2011). Directorate-General for Environment (European Commission). Recommendations for establishing Action Programmes under Directive 91/676/EEC concerning the protection of waters against pollution caused by nitrates from agricultural sources. Part A. Appendix 2, Maps of pedo-climatic zones in Europe. (2011). <https://publications.europa.eu/s/m14g>.

-256: "Our results call for a paradigm shift in vulnerability assessments": is this really a paradigm shift? Farmers have been knowing for long time that there are periods (typically wet winters) when fertilizations should not take place.

Response: With the "paradigm shift", we referred here to move a shift from currently used (static) indices based technique towards the transient vulnerability assessment. We nevertheless have revised the text – replace the wordings of "paradigm shift" to "improved assessment approaches" which better fits to the overall context of the study.

-351-353: this sentence sounds a bit vague and unjustified, so I suspect it would be better to remove it.

Response: We have followed the reviewer suggestion in this respect. Thank you.

-356-358: this is not just “worth mentioning”. This is very important and was previously shown (e.g. Remondi et al. 2018) to be a meaningful approach to simulate very non-randomly-sample dynamics.

Response: Thank you. We made a note of this important point in the revised manuscript as the reviewer suggested.

-line 94: ranges (..) increases.

Response: Thank you. We have checked the sentence as suggested.

-250-252 there seems to be something wrong with this sentence

Response: We have revised the sentence – made it concise and short so that the overall message is clear.

-equation (2): it looks similar to eq (1) so I guess there should be a minus within the exponential argument. And it looks like the variable theta is just a normalization factor in the end, so the formula might not be needed.

Response: Thank you for pointing out this typo. We have rectified this in the revised manuscript. We nevertheless keep the formulation of theta in the main text for the sake of completeness.

-323: complementary

Response: Thank you. We have rectified this typo in the revised manuscript.

-the environmental factors are sometimes indicated as f_e or as fE . Similarly, both f_s and fS are used.

Response: Thank you. We have revised the text to make the use of abbreviation consistent throughout the manuscript.

BIBLIOGRAPHY

-Sprenger M, Stumpp C, Weiler M, Aeschbach W, Allen ST, et al (2019) The demographics of water: A review of water ages in the critical zone. Rev Geophys 2018RG000633. doi: 10.1029/2018RG000633

-Remondi F, Kirchner JW, Burlando P, Fatichi S (2018) Water Flux Tracking With a Distributed Hydrological Model to Quantify Controls on the Spatiotemporal Variability of Transit Time Distributions. Water Resour Res 54:3081-3099. doi: 10.1002/2017wr021689

Response: Thank you. We have included the suggested references in the revised manuscript.

Reviewer #2 (Remarks to the Author):

In this paper, the authors use estimates of static and transient root-zone hydrologic travel times to identify the locations of nitrate-vulnerable zones across Europe. First, the authors show that humid areas with higher rainfall have shorter mean travel times. Conversely, semi-arid areas have much longer travel times and also a greater standard deviation for the daily soil-water travel times. They also show seasonal variability in travel times, with dryer summers demonstrating longer travel times. In other words, the authors show that when there is more precipitation, there is faster movement of water and solutes to the subsurface. With regard to denitrification dynamics, the authors note that denitrification rates are poorly constrained, especially at large scales. Accordingly, they do not use “real” estimates of denitrification rates, but instead take what might be considered a “toy model” approach, and consider a range of reaction time scales, i.e. how long will it take to denitrify 50% of the initial nitrate substrate? Using the relationship between the travel time and the reaction time scale, as represented by the Damköhler number, Da , the authors explore how nitrate transport from the root zone to groundwater might vary across the European landscape. Based on their coupled hydrologic-biogeochemical metric, the authors speculate that areas with faster travel times will be more vulnerable to groundwater nitrate contamination. The issue of nitrate contamination of groundwater is an important one. However, it is unclear whether the current manuscript represents a significant advance over current work or is reliable in its final conclusions.

We appreciate the reviewer's efforts and comments on our manuscript. Below we provide the point by point response to specific comment of the Reviewer. However in the onset, we would like to restate the main focus of our study, since we have the impression that a significant part of the reviewer's comments are caused by a misunderstanding of its scope and aims. This study is not written from the perspective of nitrate contamination in the groundwater, but rather in the root zone, which coincides with the first one meter of soil for arable land and which is a highly dynamic and complex part of the terrestrial system. We mainly focus on illustrating the unrecognized importance of the transient nature of (nitrate) vulnerability mapping at the continental scale in the context of potential leaching from the root-zone. This is mentioned right at the onset of the introductory paragraphs and this is identified as one of the novel contributions by the other reviewer (#1) “the large spatial scale and resolution of the model, which was never achieved before at the root-zone level”. We do not perform the analysis over the entire vadose zone, generally being much deeper than the active root-zone layer, and after which the groundwater appears. This distinction has been overlooked by the reviewer, who views this study from the perspective of “The issue of nitrate contamination of groundwater” and who also refers to the paper of Ascott et al. (2017), which focuses on nitrate storage below the

root zone. We realise that parts of our manuscript were not clear in this regard and we now have added further clarification on the scope of our study. In particular, we now state in the revised manuscript: "Our study therefore focuses on Europe-wide vulnerability assessment i.e., potential for (excess) nitrate leaching from root zone to deeper subsurface (i.e., vadoze zone below rooting depth)".

Finally, we do not agree with the reviewer's criticism regarding our lack of "real" estimates for the denitrification rates. In fact, such estimates are not available, and consequently we do not find any data source providing "real" (observational) estimates of denitrification rates even at a catchment or continental scale – the vast swath/domain we covered in our study. If high-quality observational data sets would become available to us, we would be delighted to incorporate them into our assessment. In our study, to compensate for this lack of data and represent the unavoidable uncertainty in the denitrification rates, we "consider a range of reaction time scales" as correctly stated by the reviewer. We carefully discuss and compare the results obtained with these different scenarios (See L. 157 ff. and Fig. 3 of the revised manuscript). Given the (data) situation, this is the best we can do and we hereby call for more research efforts in this direction.

Specific Comments:

1) The authors provide estimates of static and transient hydrologic travel times within the root zone based on model results using mHm, a meso-scale hydrologic model. The variability of these travel times across Europe is interesting and provides a useful basis for understanding solute transport across the continent. It is concerning, however, that the model only simulates natural hydrologic behaviour and does not account for major human intervention activities. Areas at the highest risk for nitrate leakage to groundwater (cropland) are also the most likely to be artificially drained. Areas with artificial subsurface drainage systems will have drastic decreases in travel times, but will also have preferential routing of high-nitrate runoff to surface water. In this regard, the modeled travel times, and the inferred risk to groundwater, will in many cases be completely unrelated to actual vulnerability. Significant amounts of cropland across Europe are artificially drained, so this lack of attention to tile-drainage effects is a significant lack in the current study. As noted in the manuscript, the model also does not take into account increased soil water content due to irrigation, which would clearly affect any quantification of travel times. Though this lack is acknowledged by the authors, such acknowledgement does not add confidence to our interpretation of the current reported results.

We appreciate that the reviewer finds our analysis of the travel times across Europe interesting and can provide "a useful basis for understanding solute transport across the continent". The reviewer have raised the concerns regarding the model simulations, and specifically on human intervention activities as

“artificial subsurface drainage systems or tile-drainage effects” and “irrigation effects”. To this end, we explain below that these effects have negligible influence on the overall findings of this study.

We conducted the Europe-wide simulations of hydrologic fluxes and storages focusing on synthesising the large-scale transport behavior and providing assessment on subsurface vulnerability to (excess) nitrate leaching through soil. Our model simulations represent hydrologic behaviors occurring within the terrestrial compartment - near and below Earth surface. First order changes in hydrological processes due to human intervention to terrestrial landscapes are implicitly/explicitly represented in the model, as for example by an explicit accounting for the space/time variability of vegetation and landcover dynamics, or implicitly in the parameterization of interflow components accounting for landscape attributes (varying slopes and overlying vegetation activities) that affect the near-surface drainage and flow components. With regard to the latter example, the model does not explicitly account for any physical intervention as for example in case of tile-drainage that promotes the transport activities. We agree with the reviewer that tile drainage will affect the overall travel time in the catchment. However, we expect that such direct interventions would have minimal effect on the overall modeling results presented in this study given the case that our assessment is focused on the transport processes within the root-zone soil, and the tile-drainage infrastructures are generally installed below the rooting depth (as e.g., detailed in the FAO Irrigation and Drainage Paper-62).

Regarding the issue on irrigation which was raised by both reviewers, we conducted a through investigation to analyse the impact of irrigation on the modeling results. In this regard, first of all we would like to point out that across the European arable landscapes, according to the European Statistics report (EUROSTAT 2016) in 2005, 10.1% of utilised agricultural area in the EU was irrigable but only 6.8% was actually irrigated; and over the time between 2005 and 2016 these have even decreased by 3.5% for the irrigable areas and irrigated areas by 6.1%. There are however regional variations and Spain (15.7%) and Italy (32.6%) have the largest shares of irrigable areas in the agricultural areas (EUROSTAT 2016). Nevertheless we also performed a controlled simulation to analyze the impact of irrigation on resulting transport behaviors. To this end, we adapt an irrigation water demand scheme of the well-established PCR-GLOBWB model (Wada et al., 2014; Sutanudjaja et al., 2018) and contrast the model simulation results of the irrigation (Irr) versus no-irrigation (No-Irr) scenarios. Our simulation results shown in Figure 5 confirm that large part of the cultivated area across Europe show minor differences in simulated hydrological fluxes (evapotranspiration ET or leached water I from the soil profile) due to irrigation activities. For example, the changes in evapotranspiration across 95% of cropland cells is less than 50 mm/y. Only few cropland cells ($\approx 1.8\%$) with heavy deployment of irrigation in regions of Spain and east European countries

exhibited changes in ET of more than 100 mm/y. Similarly the majority of cropland cells ($\approx 98\%$) show changes in I of less than 50 mm/y between Irr and No-Irr simulation runs. These simulation results of changes in ET and I between irrigation and no-irrigation runs provide plausible conditions considering that farmers will irrigate when atmospheric demand is high and as much such that plant can take up most of the (irrigated) water and relatively smaller portion will loss or drain to deeper subsurface. Next we analyze the effect of irrigation on the resulting transport dynamics and our simulation results indicated that increased water availability due to irrigation resulted in reduced transport times (Fig. 6). However, the majority of the cropland cells ($\approx 96\%$) exhibited changes in the median travel times (TT_{50}) of less than a month; and only few cropland cells ($< 1\%$) showed changes in TT_{50} of more than 2 months (Fig. 6). In terms of relative changes in TT_{50} with respect to the No-Irr scenario runs, almost 70% of the cropland cells show no changes in TT_{50} and vast majority ($\approx 95\%$) showed a minimal change ($\leq 5\%$). Notable changes ($> 10\%$) in TT_{50} due to irrigation were seen in only 0.7% of the cropland cells (Fig. 6). In conclusion, these simulation results demonstrate that irrigation activities have a very limited impact on the inferred large-scale patterns of transport dynamics conducted at the European scale. We describe these modelling results/discussions related to irrigation/no-irrigation scenarios in the Section “Data processing and model establishment” of the revised manuscript.

FAO IRRIGATION AND DRAINAGE PAPER 62: Guidelines and computer programs for the planning and design of land drainage systems <http://www.fao.org/3/a0975e/a0975e00.htm>

EUROSTAT 2016: https://ec.europa.eu/eurostat/statistics-explained/index.php/Agri-environmental_indicator_-_irrigation

Wada et al. (2014): Global modeling of withdrawal, allocation and consumptive use of surface water and groundwater resources. *Earth System Dynamics*, 5 (1), pp. 15–40.

Sutanudjaja et al. (2014): PCR-GLOBWB 2: a 5 arcmin global hydrological and water resources model. *Geoscientific Model Development*, 11 (6), pp. 2429–2453.

Figure 5: Spatial distribution of changes in average evapotranspiration (ΔET) and leached water (ΔI) from the soil profile between the model simulations of irrigation and no-irrigation scenarios. Bottom panels depict the histogram of changes in correspondingly fluxes over the cropland cells. Note that both fluxes show increase in the irrigation scenario compared to the no-irrigation scenario.

Figure 6: Spatial distribution of changes in mean hydrologic transport time scales (median travel time; ΔTT_{50}) between the model simulations of no irrigation and irrigation scenarios. Shown in the right are the relative changes in TT_{50} with respect to base estimates of the no-irrigation scenario. Bottom panels depict the corresponding histograms of changes in ΔTT_{50} over the cropland cells. Note that in this case the median travel times (TT_{50}) are reduced in case of the irrigation scenario (i.e., signs here are opposite compared to the above changes depicted in Fig. 5 for the simulated fluxes case).

2) The authors provide a map of groundwater vulnerability that is produced by coupling the calculated hydrologic travel times with a generic, moderate estimate of the denitrification timescale. The authors suggest, in the introduction, that their work can provide an integrated understanding of the complex and dynamic interplay between hydrologic transport and biogeochemical turnover. They also indicate that their RT50 metric, which is used to characterize the denitrification timescale, is an effective timescale encapsulating relevant environmental factors such as soil moisture, temperature and organic content, that affect the observed site-specific reaction cases. In reality, however, there is no site-specific estimate of denitrification. The mapped vulnerabilities to nitrate leaching provided in the paper assume an average denitrification rate constant that is uniformly applied across Europe, regardless of variations in soil moisture, temperature, and soil organic matter content. As a result, the findings of the paper are essentially based solely on the differing travel times rather than any real consideration of spatially varying biogeochemical turnover.

Unfortunately, we disagree with this comment regarding our work. Firstly, we would like to clarify that we provide a map of vulnerability to leaching from the root zone and not of groundwater vulnerability. As we already mentioned previously in our response to the reviewers, we further clarify the scope of our study in the revised manuscript. Secondly, our whole methodology for the assessment of transient vulnerability is based on the transient denitrification time scale, that explicitly considers the effect of space-time variability of the above mentioned environmental factors – i.e., soil moisture, temperature and organic content. See the main text (Line 205 ff.; Figs. 3c,d) and methodology section (Line 405 ff.). The potential denitrification time scale should not be confused with the actual space-time varying denitrification time scale, with the latter being actually used in our transient analyses.

3) The authors appear to assume that any cultivated area has uniformly available excess nitrate that may leach to groundwater under appropriate hydrologic and biogeochemical conditions. This assumption does not take into account variations in source nitrate across different crop types and cropping intensities. Nitrate availability must be as important as transport pathways and travel times when considering the vulnerability of groundwater to nitrate contamination, so this lack of consideration of sources is a significant weakness of the paper.

While technically the reviewer is correct in his/her statement, this is immaterial since this study is about **vulnerability and not risk**, as implied by the reviewer's comment. Our analysis aimed at providing a methodology for delineating the nitrate vulnerable zones following a vulnerability assessment framework taking into account the system propensity for nitrate leaching. The reviewer refers here to the risk assessment analysis which combines the vulnerability and the actual amount of excess nitrate available.

Our study here focus on characterising the vulnerability of the system - as this aspect is inherently related to a system functioning and needs to be known in detail to better inform decision making (e.g., fertilizer applications). On the other hand, the actual amount of excess nitrate availability for risk assessment at the continental scale is fraught with large uncertainty and its availability is limited to few snapshots in time. To deal with this issue, we have nevertheless attempted to confront our vulnerability assessment with a spatially-varying map of excess nitrate (one snapshot in time), and elaborated the corresponding results in the manuscript (see L. 461 ff; Supplement Fig. S12). To avoid any further misunderstanding, we have now revised the text indicating “Our results call for improved vulnerability assessment approaches in Europe and other regions of intensive agriculture. Current practices that do not consider transient dynamics could lead to a substantial underestimation of the extent of vulnerable areas and the associated risk. Our study addresses this limitation and thus provides crucial vulnerability criteria which can then be combined with information on the exposure (i.e. available data on excess nitrate components) for a proper risk assessment. To this end we provide a showcase example of vulnerability assessment combining information on excess nitrate (see Methods)”. With regard to this example analysis we mention: “Our analysis mainly considers the nitrate vulnerability across all cultivated areas without accounting for the spatial heterogeneity of N-surplus (or excess N) estimates i.e., net nitrogen balance after accounting for fertilizer and atmospheric inputs and plant uptake. Although these estimates are needed to properly characterize the risk of nitrate leaching, they are rarely available at sub-annual time-scales. Taking the example of the N-surplus balances for croplands around the year 2000 (West et al., 2014), we note that not all of the cultivated areas across the study domain have high N-surplus, especially the east European regions bordering the Black sea (see Supplement Fig. S12 for the N-surplus map and the corresponding cropland fractions in Fig. 3b). There is, however, a general tendency of increased N-surplus with the larger fraction of cropland areas (see Supplement Fig. S12). We find that around 33% of the total cultivated areas have a high N surplus ($\geq 1 \times 10^5 \text{kg-N}$) and is vulnerable to nitrate leaching under the static vulnerability assessment case (i.e., cropland cells of $D_a \leq 1$ based on the averaged TT_{50} and effective $\langle RT_{50} \rangle$ value). For the transient case based on the time-varying TT_{50} and RT_{50} , we find around 58% of the total cultivated areas with high N-surplus is temporarily vulnerable ($D_a(t) \leq 1$) to nitrate leaching for at least one-third of the year. The analysis conducted here provides a preliminary overview on the nitrate risk assessment, and further improvements in this aspect require more information on the temporal variability of N-surplus.”

West, P. C. et al (2014): Leverage points for improving global food security and the environment. *Science* 345, 325–328.

4) There is no attempt at validation of the current results. First, the mHM model is only a hydrologic model and does not include biogeochemical processes. As a result, there is no explicit modeling of stream nitrate or partitioning of nitrate fluxes between surface and subsurface pathways. An alternative approach, of course, would be to validate the toy-model predictions with actual maps of groundwater nitrate concentrations. To strengthen the current findings, I would recommend comparisons of predicted high-vulnerability areas with measured nitrate concentration values. Finally, Ascott et al. published a 2017 paper on “Global patterns of nitrate storage in the vadose zone,” in Nature Communications. While the current paper does cite the Ascott paper, it provides no discussion of how the current results correspond to the Ascott estimates of nitrate accumulation in the vadose zone. This lack of mention suggests a lack of attention to current relevant literature and also a loss of opportunity for outside validation of the current results.

While we agree that model validations against measurements should be done when possible, as stated above, this study is not about predicting groundwater concentrations. Since our focus is instead on the root zone, we have thoroughly evaluated the model predictions against (hydrologic) measurements available to us (as elaborated in details in the Supplementary materials). To the end of evaluation for nitrate leaching through the root-zone at a continental-scale, we are not aware of any dataset available at this point in time. To account for the lack of direct measurements of nitrate leaching, we performed a comparison with the leaching risk potential map provided by the EU authority, as mentioned in the current manuscript (see L. 191 ff). See also the related response to the reviewer#1's query on the comparison of our estimates and the map provided by the EU authority (see also Fig. 4 of this response document). Furthermore, we would like to mention the existing implementation of the nitrate model coupled to hydrologic modules of mHM (Yang et al., 2018) that account for detailed biogeochemical processes including in-stream nitrate and partitioning of nitrate fluxes between surface and subsurface pathways. The coupled model has been successfully validated in past studies (Yang et al 2018, 2019a,b). The formulation of reactive timescales related to denitrification process in soil used in this study follows a similar approach to that of the coupled model (i.e., a first-order denitrification in soil along with the space-time variability of environmental factors). As the reviewer #1 suggested, we have highlighted these previous successful applications of this transport model that is coupled to mHM (see L. 327 ff in the revised manuscript).

The reviewer suggests that we should compare our results to those of Ascott et al. (2017), which presumably arises from the aforementioned misunderstanding (groundwater vs. root-zone). As we noted earlier, our work focus on vulnerability of nitrate leaching through the root zone while the study of Ascott et al. (2017) deals with nitrate accumulation in the vadose zone below the root-zone

and treating the nitrate leaching through the root-zone as a source term for their modeled (vadose-zone) nitrate accumulation. Another main difference between both studies lies in the fact that the study by Ascott et al. (2017) deals with **nitrate availability (risk indicator)** under quasi steady-state conditions in vadose zone, while our study focuses on the **transient vulnerability (and not risk) assessment** of nitrate leaching through the root-zone soil layer. Because of these differences we do not think that information provided by Ascott et al. (2017) – which is valuable in its own way – can be compared with those of our study.

Yang, et al. (2018): A new fully distributed model of nitrate transport and removal at catchment scale. *Water Resour. Res.*, 54, 10.1029/2017WR022380

Yang, et al., (2019a): Sensitivity analysis of fully distributed parameterization reveals insights into heterogeneous catchment responses for water quality modeling. *Water Resour. Res.*, 55, 10.1029/2019WR025575

Yang, et al., (2019b): Autotrophic nitrate uptake in river networks: A modeling approach using continuous high-frequency data. *Water Res.*, 10.1016/j.watres.2019.02.059

REVIEWERS' COMMENTS

Reviewer #1 (Remarks to the Author):

The authors have carefully revised the manuscript and all the concerns from my previous review (which in some cases required new computations to be made) have been fully addressed. I appreciate the clarity of the response letter and the quality of the new analyses. As already stated in the first round of review, I believe that this is an excellent, innovative work that brings important implications for our understanding of vulnerability to subsurface nitrate contamination. Therefore, I recommend this paper for publication.

Reviewer #2 (Remarks to the Author):

The authors have revised their earlier manuscript, and I appreciate the responses to my review. I particularly appreciate the attempt to assess the impacts of irrigation on the modeling results.

I do, however, continue to question the authors' statement that subsurface drainage would have a negligible influence on their findings. Specifically, the authors point out that "such direct interventions would have minimal effect on the overall modeling results presented in this study given the case that our assessment is focused on the transport processes within the root-zone soil, and the tile-drainage infrastructures are generally installed below the rooting depth." While drainage is installed below the root zone, the effect of this drainage is to remove excess water from the root zone as quickly as possible so that agronomic operations are not impeded in the spring and fall. Accordingly, tile drainage has a fundamental impact on soil water storage and thus on root-zone transport, and it seems a significant gap that potential impacts of artificial drainage are not even mentioned in the main text of the paper.

Reviewers' Comments/Responses.

Reviewer #1 (Remarks to the Author):

The authors have carefully revised the manuscript and all the concerns from my previous review (which in some cases required new computations to be made) have been fully addressed. I appreciate the clarity of the response letter and the quality of the new analyses. As already stated in the first round of review, I believe that this is an excellent, innovative work that brings important implications for our understanding of vulnerability to subsurface nitrate contamination. Therefore, I recommend this paper for publication.

Response: We are thankful to the Reviewer for his/her positive assessment and support for publication of our work.

Reviewer #2 (Remarks to the Author):

The authors have revised their earlier manuscript, and I appreciate the responses to my review. I particularly appreciate the attempt to assess the impacts of irrigation on the modeling results. I do, however, continue to question the authors' statement that subsurface drainage would have a negligible influence on their findings. Specifically, the authors point out that "such direct interventions would have minimal effect on the overall modeling results presented in this study given the case that our assessment is focused on the transport processes within the root-zone soil, and the tile-drainage infrastructures are generally installed below the rooting depth." While drainage is installed below the root zone, the effect of this drainage is to remove excess water from the root zone as quickly as possible so that agronomic operations are not impeded in the spring and fall. Accordingly, tile drainage has a fundamental impact on soil water storage and thus on root-zone transport, and it seems a significant gap that potential impacts of artificial drainage are not even mentioned in the main text of the paper.

We appreciate the reviewer's remark on the artificial drainage issues. In this respect, we have revised the text including the following sentences:

"... We emphasise that our study focuses on providing a general framework for the characterization of sub-surface nitrate vulnerability and applied it to the pan-European landscape. This framework can also be applied/expanded to finer scales (e.g., catchment scale), incorporating more detailed datasets²⁷ and relevant processes to provide crucial insights into local vulnerability and assist policy intervention strategies. Examples are the localized effects of irrigation (see Supplementary Note 1 for details) and artificial (tile) drainage which can impact soil-water storage and indirectly the root-zone transport dynamics."

Editor's Suggestions

We have implemented all the suggestions as provided by the Editor (annotated in the pdf). These include (see detail changes in the revised version of the manuscript with track changes):

1. We add the section headings (Abstract, Results and Discussion).
2. We added a single sentence summary at the end of Introduction.
3. Suggestions regarding referencing to Supplementary Information (Note, Figures, Table) were implemented to conform with the Nature style guidelines.
4. All figure were modified taking into account for a colorblind-friendly palette; and made sure that every element of each figure is editable.
5. The title of the Supplementary Information was updated as suggested to Supplementary Information for "Strong hydroclimatic controls on vulnerability to subsurface nitrate contamination across Europe" by Kumar et al.
6. The video file has been removed from the Supplementary Information and it is now provided as an additional supplementary file.

Editorial Requests

Please ensure that an updated editorial policy checklist that verifies compliance with all required editorial policies is completed and uploaded with the revised article. . . .

Done.

Please check whether your manuscript or Supplementary Information contain third-party images, such as figures from the literature, stock photos, clip art or commercial satellite and map data. . . .

Done.

Data availability statements and data citations policy: All Nature Communications manuscripts must include a section titled "Data Availability" as a separate section after the Methods section but before the References. . . .

Done.

DATA SOURCES: Nature Research policies strongly encourage deposition of research data in public repositories. . . .

Done.

We are committed to ensuring clarity and avoiding ambiguity in the mathematics in our papers. Consequently, please make sure that mathematical terms throughout your manuscript. . . .

Done.

Your paper will be accompanied by a two-sentence editor's summary, of between 250-300 characters, when it is published on our homepage. Could you please approve the draft summary below or provide us with a suitably edited version.

We have revised the text as:

“Excess fertilizer use and other practices causes subsurface contamination. Here the authors conduct an assessment of water quality vulnerability across Europe, finding that 75% of agricultural regions are susceptible to nitrate contamination for least one-third of the year, two times more than using standard estimation procedure.”